# LATTS: LAtent space Test Time Scaling for diffusion language models

## Abstract

Test-time scaling (TTS) improves the performance of autoregressive (AR) large language models by adding computation at inference. While the prominent sequential TTS enhances accuracy by inducing models to generate longer chain-of-thought (CoT) reasoning, its computational overhead emerges as a drawback. Meanwhile, diffusion large language models (DLLMs) have emerged as a promising alternative that offers parallel decoding and self-correction capabilities. However, existing sequential TTS methods are incompatible with modern masked DLLMs. This incompatibility arises from two fundamental constraints: (1) standard DLLMs operate holistically on fixed-length sequences, preventing the dynamic token-level expansion required for CoT without specific training, and (2) the intrinsic coupling between refinement (i.e., denoising) steps and sequence length in standard DLLM formulations, restricting effective extension without delicate designs. We introduce LATTS, a novel sequential TTS method for DLLMs that addresses the above challenges by operating in the latent embedding space. LATTS reframes CoT reasoning from a *spatial* process of extending sequence length to a *temporal* process that uses additional computation to extend the iterative self-refinement steps over the entire sequence's latent representation. Our evaluation on the LLaDA-Instruct model shows that: with a brief post-training phase, LATTS achieves notable improvements over SFT baselines on reasoning and code generation benchmarks with gains of +4.1% on GSM8K, +4.8% on MATH, +3.2% on MBPP, and an average of +4.6% on commonsense reasoning tasks with minimal additional inference computation. These results establish sequential TTS as a promising technique for optimizing DLLMs.

## 1 Introduction

Test-time scaling (TTS) methods improve the reasoning performance of large language models by investing more computation at inference time (Zhang et al., 2025; Wang et al., 2022; Snell et al., 2024; Brown et al., 2024; Muennighoff et al., 2025). Existing TTS strategies fall into two primary categories. *Parallel scaling* techniques explore multiple decoding paths simultaneously to improve output correctness (Wang et al., 2022; Lightman et al., 2023; Renze, 2024; Brown et al., 2024). In contrast, *sequential scaling* methods guide the model to produce extended reasoning steps, known as Chain of Thought (CoT), to approach a more precise final answer (Madaan et al., 2023; Snell et al., 2024; Chen et al., 2025; Xu et al., 2025).

Meanwhile, diffusion large language models (DLLMs) have emerged as a powerful alternative to the prevalent autoregressive (AR) models (Ye et al., 2025; Nie et al., 2025; Cheng et al., 2025; Tae et al., 2025). Unlike AR models that generate tokens sequentially, DLLMs operate holistically on the entire sequence. They leverage bidirectional attention to incorporate full context to iteratively denoise the sequence to form a coherent and contextually relevant output. In particular, masked DLLMs based on discrete absorbing noise have achieved performance comparable to AR models (Nie et al., 2025; Ye et al., 2025; Gong et al., 2024).

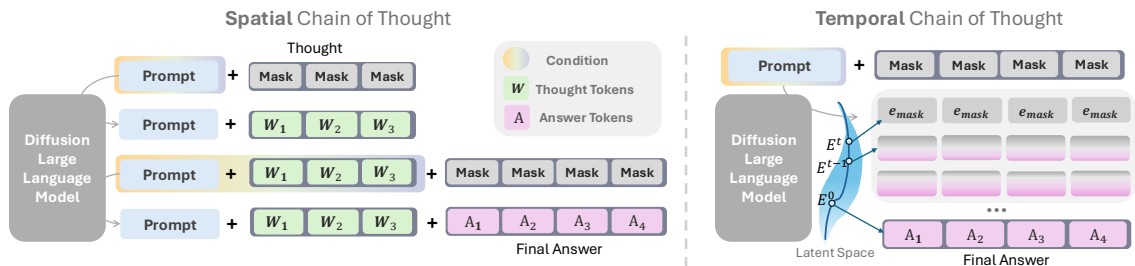

Figure 1: Different Chain of thought paradigms.

Moreover, DLLMs offer benefits such as faster decoding (Labs et al., 2025; Song et al., 2025), inherent self-refinement capabilities (Wang et al., 2025b), and unified cross-modality modeling (Yang et al., 2025a; You et al., 2025), making them a promising area for research.

Despite their potential, TTS methods tailored for masked DLLMs remain underexplored. Applying the proved successful CoT-based sequential TTS from AR models to DLLMs presents the following fundamental challenges. First, many DLLMs (e.g., LLaDA) model the joint likelihood of the full answer, which requires a predefined sequence length. Unless specifically trained (Yang et al., 2025b), this static-length constraint is incompatible with the dynamic expansion needed for CoT reasoning. Second, while DLLMs allow for a tunable computational budget via the number of denoising steps, how to effectively decouple the number of refinement iterations from the target sequence length, thereby enabling further quality improvements through additional computation, remains an open research question.

To address these challenges, we propose **LATTS** (**LAtent Space Test-Time Scaling**), a sequential TTS method that enables DLLMs to perform extended refinement. As illustrated in Figure 1, a key insight is to reframe CoT from a *spatial* process of sequential thought-token generation to a *temporal* process of iterative refinement in latent space.

The conventional approach, which we term *Spatial* CoT in this paper, mimics AR generation: it produces intermediate thoughts sequentially, where each new thought is conditioned on the previous one. This paradigm not only requires specialized training on CoT datasets (Ye et al., 2024) but also remains vulnerable to error propagation, a known weakness of sequential generation (Wu et al., 2025; He et al., 2025b), because the diffusion steps are applied to each thought rather than the entire reasoning trajectory.

We conclude the primary contributions of this work as threefold.

- We propose the *Temporal* CoT paradigm that frames reasoning as an optimization trajectory in a latent space. The core idea is to apply test-time computation to extend the refinement process along the temporal dimension, operating holistically over the entire reasoning sequence.

- Following the *Temporal* CoT formulation, we propose LATTS, a novel sequential TTS method which yields fewer cascading errors and a higher final accuracy, as evidenced by our experimental results.

- We demonstrate that LATTS is highly efficient and generalizable. It requires only a brief post-training phase and achieves enhancements across multiple downstream tasks. Our experiment shows that LATTS significantly surpasses the SFT baseline with minimal additional inference time computation. Moreover, the employment of LATTS yields reasoning performance comparable or superior to LLaDA-1.5 (Zhu et al., 2025a) trained by a DPO variant, with a drastic reduction ($> 99\%$) on training data.

## 2 RELATED WORK

**Diffusion Language Models.** Large language models (LLMs) are conditional generative models trained to approximate a data distribution $p_\theta(\cdot) \sim p_{\text{data}}$. During inference, they generate samples by maximizing the conditional likelihood given a prompt $\mathbf{c}$. Mainstream autoregressive (AR) LLMs factorize this likelihood as $\prod_{i=1}^n p_\theta(x^1|\mathbf{c}) p_\theta(x^2|x^1, \mathbf{c}) \cdots p_\theta(x^n|x^{1\ldots n-1}, \mathbf{c})$, generating tokens sequentially. In contrast, diffusion large language models (DLLMs) adapt diffusion probabilistic models (Sohl-Dickstein et al., 2015; Ho et al., 2020; Rombach et al., 2022) from continuous domains to discrete text, using a multi-step generative process to optimize the likelihood via a variational lower bound (Appendix A). Rather than continuous noise (Gulrajani & Hashimoto, 2023; Tae et al., 2025), this work focuses on discrete DLLMs with absorbing-state noise (Lou et al., 2023), also known as Masked Diffusion Language Models (MDLM). MDLMs (Austin et al., 2021a; Sahoo et al., 2024; Shi et al., 2024; Ou et al., 2024) include models initialized by continued pre-training from existing AR models (e.g., Dream (Ye et al., 2025; Xie et al., 2025), DiffuLlama (Gong et al., 2024), and SDAR (Cheng et al., 2025)) and those trained from scratch (e.g., LLaDA (Nie et al., 2025)), both achieving performance competitive with AR models. Post-training alignment further improves their reasoning. Some methods (e.g., MDPO (He et al., 2025a) and LLaDA-1.5 (Zhu et al., 2025a)) adapt DPO (Rafailov et al., 2023) to MDLMs, while others (e.g., d1 (Zhao et al., 2025), TraDo (Wang et al., 2025c), SEED-Diffusion (Song et al., 2025) and DiffuCoder (Gong et al., 2025)) apply reinforcement learning (Shao et al., 2024) to guide the denoising process.

**Test-Time scaling methods.** Test-time scaling (TTS) is an effective technique for enhancing the performance of generative models. In continuous diffusion models, parallel TTS methods such as (Ma et al., 2025) explore multiple noise predictions at each denoising step to identify higher-quality generative paths. Similarly, recent work (Wang et al., 2025b) on temporal semantic entropy extends the concept of self-consistency (Wang et al., 2022) to MDLMs. This method identifies inconsistency in intermediate answers across denoising steps and applies majority voting over step-wise predictions to enhance the stability and accuracy of the final output. Previously, ReMDM (Wang et al., 2025a) prolonged the denoising steps through strategic remasking, while ALMD (Rout et al., 2025) extended this approach by first decoding the most critical tokens and then iteratively refining the results via remasking. On the other hand, methods such as DoT (Ye et al., 2024) sequentially scale reasoning by applying the continuous diffusion process over each thought, similar to the spatial CoT in Figure 1. Recent work explores latent-space TTS methods in AR models to improve efficiency. SoftCoT++ (Xu et al., 2025) conducts parallel search from specialized initial latent tokens (similar to (Hao et al., 2024)), and LatentSeek (Li et al., 2025) optimizes latent representations directly using policy gradients. SoT (Ning et al., 2023) enhances TTS efficiency through a careful prompting design that guides the AR models to first generate a draft and subsequently refine in parallel.

## 3 PRELIMINARIES

### 3.1 MASKED DIFFUSION LANGUAGE MODEL FORMULATION

Masked Diffusion Language Models (MDLMs) consist of a dual process upon a series of intermediate states $\mathbf{x}_t$ where $t \in [0, 1]$. The forward process corrupts an initial text sequence $\mathbf{x}_0$ by progressively replacing tokens with mask tokens $\mathbf{M}$ until fully masked at $t = 1$. The reverse (denoising) process, modeled by $p_\theta(\cdot)$, iteratively fills these masks back with tokens based on a remasking strategy until no masks remain. The decoding process in a standard MDLM first initializes a sequence of $N$ mask tokens given a prompt $\mathbf{c}$. Let $\phi : \mathcal{V} \to \mathbb{R}^d$ denote the model's token embedding function, where $d$ is the embedding dimension. The initial latent state of the answer is represented as the matrix: $\mathbf{E}^{(1)} = [\mathbf{e}_{\text{mask}}, \mathbf{e}_{\text{mask}}, ..., \mathbf{e}_{\text{mask}}], \mathbf{e}_{\text{mask}} = \phi(\mathbf{M})$. The model then repeats through the following two steps until $t = 0$:

1. **Denoising Step**: It samples a fully unmasked (but potentially noisy) sequence $\mathbf{x}_{t-\Delta t} \sim p_\theta(\cdot | \mathbf{E}^{(t)}, \mathbf{c})$.

Figure 2: **LATTS solution overview.** – We plot the word embedding matrix $\mathbf{E}^{(t)}$ along the denoising trajectory. A potential energy landscape is constructed using the average final embeddings from correct trajectories as the energy minimum, with energy defined as the Euclidean distance to this reference. LATTS provides a starting point closer to the correct solution in the embedding space.

2. **Remasking Step**: It applies a remasking function $\bar{\mathbf{x}}_{t-\Delta t} = \gamma(\mathbf{x}_{t-\Delta t})$, and updates the word embedding matrix $\mathbf{E}^{(t-\Delta t)} = \phi(\bar{\mathbf{x}}_{t-\Delta t})$. For instance, under a confidence-based strategy, only tokens in $\mathbf{x}_{t-\Delta t}$ with top-$k$ confidence are unmasked; others are remasked:

$$\bar{x}_{t-\Delta t}^{i} = \begin{cases} x_{t-\Delta t}^{i} & \text{if } p_\theta(x_{t-\Delta t}^{i}|\mathbf{E}^{(t)}, \mathbf{c}) \in \max_k\{p_\theta(x_{t-\Delta t}^{j}|\mathbf{E}^{(t)}, \mathbf{c}) \mid \bar{x}_t^{i} = \mathbf{M}\} \\ \bar{x}_t^{i} & \text{if } \bar{x}_t^{i} \neq \mathbf{M} \\ \mathbf{M} & \text{otherwise} \end{cases} \tag{1}$$

Here $\Delta t = \frac{k}{N}$ thus $\bar{\mathbf{x}}_0$ is fully unmasked.

### 3.2 TOKEN UNMASK IN MDLMS AS LATENT SPACE OPTIMIZATION

Notice that the iterative unmasking procedure in MDLM can be projected into a continuous optimization process in the latent embedding space, conditioned on the prompt. Specifically, on the embedding manifold, those lead to correct answers correspond to lower-energy regions. The DLLM process starts from a uniform initial latent representation (all mask tokens) and performs optimization on the energy surface of the latent representation through iterative refinements. Standard MDLM (e.g., LLaDA) employs a fixed number of denoising steps bounded by a predefined answer length. Thus, they produce a fixed-length optimization path $\{\mathbf{E}^{(1)}, \mathbf{E}^{(1-\Delta t)}, \dots, \mathbf{E}^{(0)}\}$ where the number of unmasked tokens per step determines the optimization step size. In the above context, applying *parallel* TTS techniques equals exploring multiple optimization trajectories. With temperature $T \geq 1$, they inject noise to promote exploration of diverse local minima.

## 4 LATENT SPACE TEST TIME SCALING

As formulated in Section 3.2, the decoding process of MDLMs can be interpreted as an optimization procedure in the latent space. However, the starting point of this optimization process is fixed in the embedding space across different questions. It carries no prior information, with varying conditions only reflected in the direction of descent. Moreover, the number of optimization steps is bounded by the pre-defined target length, which may cause the trajectory to converge to a local minimum even when parallel TTS is employed.

To address this limitation, we propose LATTS, a method that leverages test-time computation to guide the model to a superior starting point in the latent space. LATTS first generates a self-sampled draft, effectively using the model's own knowledge to create an informed initialization. Consequently, this better starting point facilitates convergence to higher-quality answers. This procedure can be interpreted as an extension of the refinement process, akin to prolonging the temporal CoT as introduced in Section 1.

## 4.1 DESIGN OVERVIEW FOR LATTS

As shown in Figure 2, the LATTS framework has two key designs. First, we utilize additional computation to extract distinct priors for different prompts. Specifically, we guide the model to retrieve its internal knowledge most relevant to a given condition, which is then fused into a latent vector with semantic information. Second, the model is trained to perform iterative refinement starting from this semantically-rich latent vector. We teach the model to utilize this vector as the initial point for latent space optimization, ensuring a more guided and effective search. We introduce these two parts in Section 4.2 and Section 4.3.

The LATTS decoding process consists of two distinct phases. In Phase 1, a semantic latent starting point is constructed. Specifically, a draft answer is sampled from the model and then mapped back into the latent space. Through aggregation and extension operations, it is expanded into a latent vector of answer length. In Phase 2, the actual decoding begins. Starting with a fully masked sequence of the target answer length, the mask positions corresponding to the answer are replaced with the semantic latent vector obtained in Phase 1. The decoding then proceeds according to Equation (1). At each denoising step, the remaining mask positions are again replaced with the semantic latent vector. This iterative process continues until all masks have been replaced by actual tokens. We conducted a detailed pseudo-code in Appendix F.

## 4.2 OBTAINING LATENT VECTORS WITH RICH SEMANTIC PRIOR

Our objective is to derive semantic latent vectors from the model's internal representations that are most relevant to the given prompts. A key difficulty involves crafting a representation that possesses detailed semantics yet remains sufficiently abstract. An overly converged vector, akin to a point residing at a low-energy basin on the optimization landscape, would confine subsequent refinement steps to a local minimum, preventing meaningful refinement in later steps. This is empirically evidenced when using the converged final output as an informed initialization (as in Section 5.2.1): the model often reproduces the same tokens, failing to escape the suboptimal region.

Our proposed method begins by sampling a short draft generation from the model to serve as a semantic prior. Specifically, given a prompt $\mathbf{c}$ and a target length $N$, we first generate a draft answer $\mathbf{A}_{\text{draft}} = (w_1, w_2, \ldots, w_n)$, typically under a short-sequence constraint ($n < N$). This draft is mapped into its embedding representation $\mathbf{E}_{\text{draft}} = [\phi(w_1), \phi(w_2), ..., \phi(w_n)]$. Subsequently, we apply an aggregation function $\mathcal{A}(\cdot) : \mathbb{R}^{n \times d} \to \mathbb{R}^{M \times d}, M < n$ to $\mathbf{E}_{\text{draft}}$ to distill the semantic information into a more compact and effective latent representation, thus reducing the suboptimal region problem. The specific designs of $\mathcal{A}(\cdot)$ are elaborated below.

**Average embedding Aggregator** $\mathcal{A}_{\text{avg}}$ performs a chunk-level average among the draft embeddings. $\mathbf{E}_{\text{draft}}$ is partitioned into $M$ contiguous blocks of equal length $L = n/M$. Let the $j$-th block contain the embeddings from index $i = (j-1)L + 1$ to $i = jL$. $\mathcal{A}_{\text{avg}}$ calculates $\mathbf{a}_j = \frac{1}{L} \sum_{i=(j-1)L+1}^{jL} \phi(w_i), j = 1, 2, ...M$ and yields a compressed representation of the draft generation. Beyond fixed-length approaches, variable-length semantic chunking methods (Zhao et al., 2024; Zhu et al., 2025b; Duarte et al., 2024; Sheng et al., 2025), such as segmentation at natural punctuation boundaries or identifying cut points through token perplexity or similarity minimization, can prevent truncation of draft latent embeddings.

**1d-convolution embedding Aggregator** $\mathcal{A}_{\text{conv}}$ adopts a 1d-convolution kernel to blend the content of the draft generation. The kernel weight is initialized using a Gaussian distribution. Given a kernel size $k$, convolution is applied iteratively with a stride of $\frac{k}{2}$. The number of vectors after each aggregation step is given by $o_{i-1} = 2\lfloor \frac{o_i}{k} \rfloor - 1$, where $o_0 = N$. This process repeats until $o_i \leq M$.

### 4.3 BOOSTING MODEL REASONING WITH SEMANTIC LATENT VECTORS

This section elaborates on how LATTS leverages the semantic latent vectors to enable more effective refinement by transforming them into superior initial points within the latent space.

The core idea of LATTS is to leverage an extension function $\mathcal{X}(\cdot)$ that expands and distributes the semantic information from the aggregated draft embeddings $\{\mathbf{a}_i\}_{i=1}^{M}$ across the fully masked initial embedding matrix $\mathbf{E}^{(1)}$ of the final answer. This yields a semantically informed initial matrix $\mathbf{E}_{\text{new}}^{(1)} = \{\hat{\mathbf{e}}_1, \hat{\mathbf{e}}_2, ..., \hat{\mathbf{e}}_N\}$, effectively shifting the optimization starting point closer to a low-energy region. $\mathbf{E}_{\text{new}}^{(1)}$ is used to replace the fully masked initial matrix $\mathbf{E}^{(1)}$ in Equation (1) and the denoising process works similarly as before: the corresponding positions in the matrix are updated by the actual word embedding of the unmasked tokens:

$$\mathbf{E}_{\text{new}}^{(t-\Delta t)}[i] = \begin{cases} \mathbf{E}_{\text{new}}^{(t)}[i] & \text{if } \bar{x}_{t-\Delta t}^i = \mathbf{M} \\ \phi(\bar{x}_{t-\Delta t}^i) & \text{otherwise} \end{cases} \tag{2}$$

Replacing the initial mask embeddings of the final answer with semantic-rich embeddings offers two key advantages. First, semantic latent vectors provide additional sequence-level information, offering highly relevant information to guide the conditional generation process. Second, and more subtly, these embeddings implicitly encode token-wise similarities. To be specific, during self-attention computation, word embeddings are projected into queries, keys, and values. Each position in $\mathbf{E}_{\text{new}}^{(t)}$ acts as a query that naturally attends to semantically related keys, thereby steering the model towards more effective information integration in deeper layers. Comparative experiments in Section 5.2.1 validate that using the semantic latent vector to initialize the optimization process (i.e., in-place replace the uniform masked embeddings) constitutes an effective strategy for utilizing the model's self-sampled information for guidance.

LATTS currently adopts a block-wise duplication strategy for the extension function that produces the initial matrix $\mathbf{E}_{\text{new}}^{(1)}$, wherein each latent vector $\mathbf{a}_j$ is replicated consecutively $L' = \lfloor \frac{N}{M} \rfloor$ times: $\tilde{\mathbf{e}}_i = \mathbf{a}_{\lceil i/L' \rceil}$ where $i \leq L' \times M$, and the remaining positions are padded with the mask token embedding $\mathbf{e}_{\text{mask}}$. We then mix it with mask embedding by $\hat{\mathbf{e}}_i = \alpha \cdot \tilde{\mathbf{e}}_i + (1 - \alpha)\mathbf{e}_{\text{mask}}$ where typically $\alpha = \frac{n}{N}$.

We justify the design of preserving a portion of masked embedding in the derived new initial point for two reasons. Firstly, the model is trained under a denoising objective that primarily learns to convert mask tokens into meaningful representations. Thus, retaining a portion of the mask embedding leverages this strong inductive bias. Second, this prevents over-concentration of attention on the repeated draft representations ($\{\hat{\mathbf{e}}_i\}_{i \in ((j-1) \cdot L', j \cdot L']}$ is the same) output by the extension function in early decoding steps, which could otherwise limit generation diversity.

We also require a post-training stage to help the model learn to utilize semantic latent vectors for efficient retrieve information. The model $p_\theta(\cdot)$ inputs a partially masked sample $\mathbf{x}_t$, replacing the word embedding of masked tokens with $\mathbf{E}_{\text{new}}^{(t)}$, and calculate the cross-entropy loss on all masked tokens:

$$\mathcal{L}(\theta) = -\mathbb{E}_{t \sim \mathcal{U}(0,1), \mathbf{x}_0, \mathbf{x}_t} \left[ \frac{1}{t} \sum_{i=1}^{N} \mathbf{1}[x_t^i = \mathbf{M}] \log p_\theta(x_0^i | \mathbf{E}_{\text{new}}^{(t)}, \mathbf{c}) \right] \tag{3}$$

$$\mathbf{E}_{\text{new}}^{(1)} = \mathcal{X}(\mathbf{E}^{(1)}, \mathcal{A}(\phi(\mathbf{A}_{\text{draft}}))),$$

in which draft answer $\mathbf{A}_{\text{draft}}$ is generated from $p_\theta(\cdot)$ with the standard denoising process.

## 5 EXPERIMENTS

### 5.1 EXPERIMENTAL SETUP

**Models and Training** We employed two base models: LLaDA 8B Base and LLaDA 8B Instruct (Nie et al., 2025) (both trained from scratch as MDLMs). All models were further finetuned for 20 epochs on the s1k dataset (Muennighoff et al., 2025), using either supervised fine-tuning (SFT) or LATTS with the training objective defined in Equation (3), with a cutoff length of 2048 and a learning rate of 1e-5. For the LLaDA Base model, training was performed on both s1k and the training set of GSM8K (Cobbe et al., 2021). Furthermore, we include LLaDA-1.5 (Zhu et al., 2025a) as a baseline (denoted as VRPO), which has undergone preference alignment with 350k pairs of data upon LLaDA 8B Instruct. LATTS defaults apply the average aggregator $\mathcal{A}_{\text{avg}}$ (refer to Section 4.2), and involves three hyper-parameters: the draft answer length $n$, the block length $L$ for average embedding aggregation, and the scaling factor $\alpha$ for blending with mask embeddings. Unless otherwise specified, the default values are set to $n = 64$, $L = 4$, and $\alpha = \frac{n}{N}$, where $N$ denotes the target answer length. The number of denoising steps used when sampling the draft answer is set to be half of the draft answer length. We involve ablation studies on the hyperparameters in Section 5.3. All experiments were conducted on 8 NVIDIA H800 GPUs. The post-training phases for SFT and LATTS took 13 and 32 H800 GPU hours, respectively.

**Datasets and Inference** Our evaluation employs datasets from three distinct categories: **(1)** Mathematical Reasoning: We choose GSM8K and MATH500 (Lightman et al., 2023; Hendrycks et al., 2021), where the model is required to generate a reasoning process followed by a final answer, which is then matched against the ground truth. **(2)** Code Generation: We evaluate on the MBPP (Austin et al., 2021b) dataset using the standardized lm_eval framework. **(3)** Commonsense Reasoning: We select ARC-Challenge (Clark et al., 2018), and HellaSwag (Zellers et al., 2019). Each question includes four candidate options. Unlike the evaluation method used in LLaDA, which samples and compares the conditional likelihood of each option given the question, we instead prompt the model to produce a reasoning process before outputting the final answer. During inference with MDLMs, three key parameters may influence performance: (1) The target answer length we test values of 256 and 512; (2) The number of denoising steps is fixed to the answer length; (3) The decoding strategy in LLaDA defaults to being semi-AR, with a block size of 32. An ablation study was conducted to analyze the impact of the latter two parameters in Section 5.3 and Appendix E.3. We adopt the lm-eval-harness (lme, 2025) framework adapted for MDLMs for evaluations.

### 5.2 MAIN RESULTS

**LATTS efficiently scales the model's performance with minimal added computation.** We present the zero-shot performance of LATTS across five tasks using several base models. We exclude direct evaluation on LLaDA-8B-Base because it lacks SFT training, which hinders proper formatting compliance, making fair comparison unfeasible. According to Table 1, by adding 32 more temporal CoT steps (i.e., the number of extra refinement steps), LATTS consistently outperforms the SFT baseline across almost all tasks. LATTS outperforms or is on par with the VRPO-tuned model (a variant of DPO) on mathematical and coding tasks. We also observe that LATTS exhibits strong generalization across diverse task domains, although the model was only trained on mathematical datasets. We hypothesize that the use of self-sampled drafts in post-training acts as a regularizer to constrain the optimization trajectory. This prevents excessive distribution shift and is likely why the model maintains its competence on commonsense tasks.

**Temporal CoT outperforms spatially extending the answer length.** Notice that LATTS with a target length of 256 achieves superior performance on tasks like MATH500 and MBPP compared to both the SFT and original models generating 512-length answers, despite requiring less computation. Given that extending answer length constitutes a form of sequential TTS, this evidence leads us to argue that latent-space iterative refinement is a more effective strategy than spatial extension in many cases.

Table 1: **Model performance on different tasks** – Green values marks the enhancement of LATTS over the SFT baseline. The amount of data used for post-training is indicated in parentheses. Best and second -best results are highlighted in **bold** and underlined, respectively.

| Model | Method | GSM8K | | MATH500 | | MBPP | | ARC-C | | Hellaswag | |
|-------|--------|-------|-----|---------|-----|------|------|-------|------|-----------|------|
| | | 256 | 512 | 256 | 512 | 256 | 512 | 256 | 512 | 256 | 512 |
| LLaDA-8B-Base | SFT (1k+8k) | 65.6 | 72.3 | 28.8 | 28.8 | **40.8** | 39.2 | 73.8 | 71.2 | 80.0 | 81.1 |
| | LATTS (1k+8k) | 73.0 | 74.0 | 31.2 | 30.2 | 40.4 | 39.2 | 80.9 | 79.8 | **81.5** | **81.4** |
| | | +7.4 | +1.7 | +2.4 | +1.4 | -0.4 | 0.0 | +7.1 | +8.6 | +1.5 | +0.3 |
| LLaDA-8B-Instruct | Direct (-) | 76.7 | 78.2 | 24.8 | 30.2 | 36.2 | 37.8 | 80.3 | 80.4 | 72.1 | 73.9 |
| | SFT(1k) | 75.4 | 82.8 | 35.6 | 37.0 | 37.8 | 36.8 | 77.4 | 74.3 | 66.3 | 66.9 |
| | LATTS (1k) | 79.5 | **84.7** | **37.2** | **41.8** | **40.8** | **40.0** | 80.9 | 79.7 | 70.4 | 72.1 |
| | | +4.1 | +1.9 | +1.6 | +4.8 | +3.0 | +3.2 | +3.5 | +5.4 | +4.1 | +5.2 |
| | VRPO(350k) | **79.8** | 83.1 | 36.0 | 37.6 | 37.8 | 38.2 | **82.9** | **83.2** | 73.3 | 75.4 |

### 5.2.1 COMPARISON WITH *Sequential* TTS BASELINES

We design five sequential TTS baselines for diffusion language models and evaluate their performance-computation trade-offs. The first category, spatial CoT, extends the answer length: (1) **Static Extend**: Start with a fixed longer answer length of $n + N$. (2) **Dynamic Extend**: During semi-AR decoding, the sequence is initialized with a fixed-length, fully masked block of size $N$. After unmasking each block, the generation process dynamically appends a fully masked block to the back of the sequence until an EOS token is generated or the total length reaches a maximum of $2N$. (3) **Concat Guidance**: Sample an $n$-token draft, concatenate it to the prompt, and use system instructions to guide the model to refine the final answer based on the draft. (4) **Concat Latent**: Similar to (3), but instead of concatenating the draft answer, we concatenate the averaged latent embedding aggregated by LATTS, which also requires an additional training process. The second category, temporal CoT, includes: (5) **Temporal Remask**: After the model generates a final answer, we perform $n$ additional denoising steps by remasking a subset of tokens with the lowest confidence and denoising them. (6) **ReMDM**: According to Wang et al. (2025a), we adopt the loop strategy after partially generating the answer. Specifically, once the model produces an initial final answer, we remask the $N/8$ tokens with the lowest confidence. The loop process is then executed for $n$ times, where in each iteration $N/64$ (i.e., 4 or 8) tokens with lowest confidence are remasked and subsequently $N/64$ masked tokens are unmasked again. Finally we unmask the rest $N/8$ masked tokens. We set the temperature to 1 in order to prevent the model from getting trapped in meaningless loops.

In Figure 3, we compare the computation-performance trade-off of the six baseline methods against LATTS. We evaluate the average inference latency of the six aforementioned methods under different settings, with a batch size of 1 per GPU and ($n = 64$). For ReMDM, we also report results with $n = 512$ for comparison.

Several key observations emerge: First, LATTS nearly achieves a Pareto improvement, delivering greater performance gains with minimal computational overhead. Second, spatial extension methods generally require substantially more computation due to longer context processing, although an exception occurs on the GSM8K task with 512 answer length, where early stopping at EOS reduces overall cost (see Appendix B, Table 2 for average answer lengths). Third, the low-confidence temporal remasking strategy yields no meaningful improvement, as remasking tokens after full decoding often leads to regenerating identical tokens. We provide statistics on the number of modified tokens per task and case studies in Appendix D Table 3. Finally, while increasing the loop count in ReMDM improves accuracy, it also raises computational cost, resulting in a less favorable efficiency–accuracy trade-off than LATTS.

| Method | GSM8K | | | | MATH500 | | | |
|---|---|---|---|---|---|---|---|---|
| | 256 | | 512 | | 256 | | 512 | |
| | Time (s) | Acc (%) | Time (s) | Acc (%) | Time (s) | Acc (%) | Time (s) | Acc (%) |
| Baseline | 10.39 | 75.4 | 24.68 | 82.8 | 10.46 | 35.6 | 24.75 | 37.0 |
| Dynamic Extend | 13.17 | 79.0 | **17.82** | 83.4 | 15.55 | 35.4 | 35.42 | 40.6 |
| Static Extend | 13.41 | 79.3 | 28.49 | 82.7 | 13.42 | 37.0 | 28.62 | 39.6 |
| Concat Latent | 11.78 | 78.0 | 26.39 | 80.3 | 11.85 | 32.8 | 26.49 | 37.2 |
| Concat Guidance | 11.93 | 71.8 | 26.8 | 80.4 | 11.92 | 32.4 | 26.57 | 36.8 |
| Remask(64) | 12.52 | 75.2 | 26.69 | 83.3 | 12.56 | 36.6 | 26.81 | 37.4 |
| ReMDM(64) | 13.71 | 76.6 | 30.04 | 82.7 | 13.90 | 34.4 | 30.42 | 40.2 |
| ReMDM(512) | 21.04 | 76.8 | 54.47 | 84.3 | 21.3 | **37.2** | 51.33 | 40.0 |
| LATTS(64) | **11.58** | **79.5** | 25.78 | **84.7** | **11.64** | **37.2** | **25.99** | **41.8** |

Figure 3: **LATTS vs. TTS baselines.** – Average latency per sample ($\downarrow$) and the accuracy ($\uparrow$) across two tasks and two answer lengths.

| pass@4 | | GSM8K | | MATH500 | |
|---|---|---|---|---|---|
| paralle TTS | Method | 256 | 512 | 256 | 512 |
| | Direct | 89.5 | 91.1 | 45.0 | 44.2 |
| Majority Vote | SFT | 89.9 | 91.4 | 49.0 | 49.0 |
| | LATTS | 91.7 | 92.0 | 48.6 | 53.0 |
| | Direct | 89.7 | 92.2 | 45.4 | 47.2 |
| Beam Search | SFT | 90.4 | 92.0 | 49.8 | 48.4 |
| | LATTS | 92.0 | 92.6 | 49.0 | 51.2 |

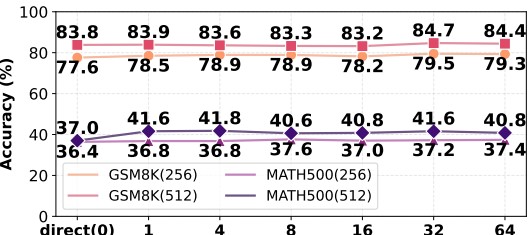

Figure 4: Combine sequential with parallel TTS

Figure 5: Scaling laws of varying #denoising steps.

### 5.2.2 COMBINED WITH *Parallel* TTS METHODS

In common practice, sequential and parallel TTS are often used in a hybrid manner. In this section, we focus on evaluating the overall performance of the proposed method when combined with parallel TTS. We implemented two variants of parallel TTS on MDLMs: **(1) Majority Vote**: Sampling is performed $k$ times with temperature set to 1. **(2) Beam Search**: We perform a beam search with beam size $k$ with a similar formulation as in *diverse beam search* (Vijayakumar et al., 2018). We also add a diversity penalty discouraging redundant continuations. Details can be found at Appendix E.1. As reported in Figure 4, we compare the pass@k (k=4) results across the following settings: the base LLaDA Instruct model, the model after SFT, and the integration of LATTS (with default settings) with each of the two parallel TTS strategies. LATTS keeps its performance advancement in most cases.

### 5.3 DESIGN SPACE OF LATTS AND ABLATION STUDIES

**Added Computation vs. Accuracy Enhancement** In this section, we discuss the scaling laws of LATTS and the trade-off space offered by its different hyperparameter choices. As shown in Figure 5, by expanding the additional computational cost via increasing the diffusion steps for the draft answer, the final answer refinement process also benefits from an improved starting point, leading to enhanced performance. In Appendix E.4, we conducted a sweep of denoising steps across three different draft answer lengths on the MBPP benchmark and observed a performance plateau in all cases. Furthermore, employing LATTS training without additional sampling at test time yields gains compared to SFT, which may be attributed to the training method resulting in a smoother embedding space. Results of varying draft answer length are reported in Appendix E.2.

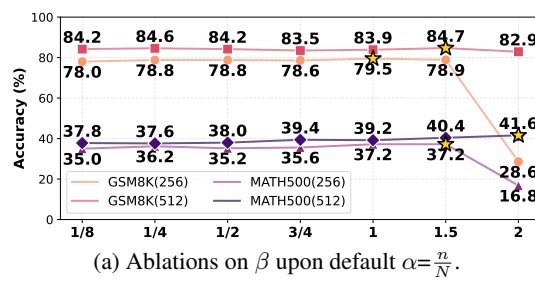

| | GSM8K(256) | (512) | MATH(256) | (512) |
|---|---|---|---|---|
| $L = 2$ | 78.9 | 83.3 | 37.2 | 40.4 |
| $L = 4$ | 79.5 | 84.7 | 37.2 | 41.8 |
| $L = 8$ | 78.5 | 83.2 | 36.6 | 37.2 |
| $\mathcal{A}_{\text{avg}}$ | 79.5 | 84.7 | 37.2 | 41.8 |
| $\mathcal{A}_{\text{conv}}$ | 78.6 | 84.2 | 37.4 | 40.6 |
| SFT | 75.4 | 82.8 | 35.6 | 37.0 |

(a) Ablations on $\beta$ upon default $\alpha = \frac{n}{N}$.

(b) Ablations on Aggregators $\mathcal{A}$ and context length $L$.

Figure 6: Ablation studies on LATTS hyperparameters.

| Model | Method | GSM8K | | MATH500 | | MBPP | | TruthfulQA | |
|---|---|---|---|---|---|---|---|---|---|
| | | 256 | 512 | 256 | 512 | 256 | 512 | 256 | 512 |
| LLaDA-2B | Direct | 55.4 | 51.5 | 14.0 | 12.6 | 27.2 | 26.6 | 13.8 | 18.1 |
| | SFT | 56.3 | 49.8 | 14.2 | 12.6 | 27.0 | 26.6 | 19.7 | 24.7 |
| | LATTS | **58.2** | **52.6** | **14.8** | **14.0** | **27.4** | 26.4 | **21.3** | **25.4** |
| | | +1.7 | +2.8 | +0.6 | +1.4 | +0.4 | -0.2 | +2.6 | +0.7 |

Figure 7: Ablation studies on Model Size.

**Hyper-parameter tradeoff of LATTS** Under a fixed computational budget, several primary hyperparameters influence the performance of LATTS. To investigate the mixing ratio $\alpha$ (default $\frac{n}{N}$) between the aggregated embedding and the mask, we scale the default value $\alpha = \frac{n}{N} \cdot \beta$ and vary the scaling factor $\beta \in [\frac{1}{8}, 2]$. In Figure 6a, an optimal value of $\beta$ exists across different tasks and answer lengths. The best-performing $\beta$ increases with problem difficulty, suggesting that harder tasks benefit more from a stronger prior via a superior starting point. Results in Figure 6b indicate that a moderate window size for mean pooling in $\mathcal{A}_{\text{avg}}$ yields the best performance. This is because a larger window size may cause information distortion through excessive smoothing, while a smaller window fails to capture sufficient contextual information. $\mathcal{A}_{\text{conv}}$ (kernel size=4, stride=2) yields comparable performance to $\mathcal{A}_{\text{avg}}$, both exceeding the SFT baseline.

**Different Model Size** To investigate the performance of LATTS across different model scales, we trained a 2B-parameter LLaDA model. The training phase consisted of 1.1T tokens for pre-training followed by 4B tokens for SFT. Subsequently, we use the s1K dataset to post-train the model for 10 epochs, applying both standard SFT and the LATTS method, with all other settings consistent with Section 5.1. As shown in Figure 7, LATTS maintains its performance advantage across various tasks. However, likely due to the more limited capacity of the 2B model, the generated draft answers may contain less informative content, resulting in a less pronounced absolute improvement gain compared to the 8B model.

## 6 CONCLUSION

Applying sequential test-time scaling to discrete diffusion language models has been underexplored. We highlight the limitations of directly applying spatial CoT to Diffusion Large Language Models and introduce a novel direction, *Temporal* CoT, which reallocates computational resources toward increasing the number of refinement steps. We present LATTS, where the model self-samples to identify initial latent points that converge toward correct answers with higher probabilities. Extensive experiments demonstrate that performing TTS along this temporal dimension efficiently enhances MDLM's performance. Future work may further advance this paradigm by reducing the overhead of the draft answer generation, employing smaller models to predict latent vectors, investigating semantic-based chunking aggregators, and exploring more efficient latent space optimization strategies.

REPRODUCIBILITY STATEMENT

In this work, we introduce LATTS, a novel sequential test-time scaling method for Diffusion Large Language Models (DLLM). To ensure that our contributions can be easily reproduced and extended by the research community, we provide source code implementing all components of LATTS. The package includes modules for draft answer generation, latent embedding aggregation, embedding extension, and temporal refinement during inference and post-training. Experimental settings and hyperparameters are reported in our code repository.

We provide open-source code to reproduce our experiments in the following anonymous repository: https://anonymous.4open.science/r/LATTS-AF45.

ETHICS STATEMENT

This work adheres to the ethical guidelines set forth by ICLR 2026. We have conducted our research with a commitment to avoiding harm, ensuring honesty and transparency in our methodology and reporting. We have made concerted efforts to identify and mitigate potential biases in our data and algorithms to ensure fairness. Furthermore, our research respects individual privacy, and we have complied with all applicable regulations and ethical standards regarding data use.

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

## THE USE OF LLMS

This paper employed LLMs only for language polishing in parts of the text.

## APPENDIX A  FORMULATION OF MASKED DIFFUSION LANGUAGE MODELS

Masked diffusion language models train on a sequence of tokens $\mathbf{x}_t = (x_t^1, \ldots, x_t^n)$ parameterized by $t \in [0, 1]$. The forward diffusion process $q$ is defined by $\mathbf{x}_0$ at $t = 0$ and proceeds by progressively replacing tokens with a special mask token $\mathbf{M}$. Hence, $\mathbf{x}_t$ corresponds to a sequence with an expected masking ratio that increases with $t$. Each token follows the conditional distribution

$$q_t(x_t^i \mid x_0^i) = (1 - t)\,\delta(x_t^i = x_0^i) + t\,\delta(x_t^i = \mathbf{M}),$$

where $t \in [0, 1]$ denotes the masking rate and $\delta(\cdot)$ is the Dirac measure while guaranteeing that $x_s^i \neq \mathbf{M}$ or $x_t^i = \mathbf{M}$ for any $0 \leq s < t \leq 1$ and any $i = 1, 2, \ldots, n$. Intuitively, each token is masked at some time in $[0, 1]$ by the masking ratio $t$ and is guaranteed to be masked at the end of the forward process, i.e. $t = 1$.

The model $f_\theta(\cdot \mid \mathbf{x}_t)$ predicts all masked tokens in $\mathbf{x}_t$ simultaneously, i.e. learns $q_t(\mathbf{x}_0 \mid \mathbf{x}_t)$. Training is performed by minimizing a NELBO on the log-likelihood:

$$\text{NELBO}(\theta) \;=\; \mathbb{E}_{\mathbf{x}_0 \sim p_{\text{data}},\, \mathbf{x}_t \sim q_t(\cdot | \mathbf{x}_0)} \left[ -\int_0^1 \frac{\mathrm{d}t}{t} \sum_{i=1}^{L} \mathbf{1}[x_t^i = \mathbf{M}] \, \log f_\theta(x_0^i \mid \mathbf{x}_t) \right],$$

where $\mathbf{1}[x_t^i = \mathbf{M}]$ equals to 1 if $x_t^i = \mathbf{M}$ and equals to 0 otherwise. Thus, the model learns to invert the forward process by denoising (unmasking) the masked sequence.

## APPENDIX B  ELABORATION ON SEQUENTIAL TTS BASELINES

This section elaborates on the implementation and training details of different sequential TTS baseline methods. With the exception of Concat Latent, which was tested on a specifically trained model, all other methods were applied to the SFT-finetuned model.

**Dynamic Extend**: The model adopts a semi-AR remasking strategy, and tokens are unmasked from left to right. Unmasked tokens are treated as prompts; once generated, they are cached, and attention is no longer computed upon them.

**Concat Guidance** and **Concat Latent**: the prompt of these two methods consists of: **(1) general system prompt** (e.g. "You will be given a question to solve. Solve it step by step. First output the reasoning process and wrap the final answer in a boxed.") **(2) an additional system prompt** steering the model to refine upon the draft answer or latent vector: "Please refer to the following draft answer and output a more complete and refined answer step by step. Draft answer:". **(3) draft answer or aggregated embeddings**.

**Concat Latent** aggregates draft answers in a manner similar to LATTS and concatenates the non-extended latent vectors to the prompt. It thus requires training analogous to LATTS. Specifically, during the model's forward pass in training, we attach placeholder tokens at positions of (3) and replace the hidden states of these placeholder tokens with the aggregated embeddings.

**Temporal Remasking**: We randomly selected $n$ positions for remasking. Subsequently, a high-confidence remasking strategy was employed to unmask one token per step.

### B.1  DETAILS OF APPROXIMATING THE TOTAL COMPUTATION OVERHEAD

This section outlines the computational complexity analysis for the evaluated baseline methods. The analysis focuses on the dominant cost: the self-attention mechanism, which scales as $O(n^2 d)$ for sequence length $n$ and hidden dimension $d$.

A key constraint is that DLLMs employ full attention, which inherently lacks support for key-value (KV) caching during the generation phase. For simplicity, however, we assume the prompts are cached after the prefill stage. Consequently, the attention computation cost associated with the prompt length is excluded for all methods. The primary computational overhead is thus attributed to the denoising steps.

During each denoising step, the model must compute full self-attention over the entire current context length (equal to the pre-defined answer length $N$). Since our default setup in LLaDA utilizes semi-ar generation, the process terminates upon generating an end-of-sequence (EOS) token. If one token is unmasked per step, the number of denoising steps equals the effective generated length (i.e., the sequence length before EOS).

Therefore, we model the total complexity $T_{\text{total}}$ of the diffusion process as the number of denoising steps $S$ multiplied by the square of the context length $N_{\text{context}}$, formally: $T_{\text{total}} = S \cdot N_{\text{context}}^2$.

Assume the final answer length $N$ and the draft answer length $n$. Average answer length denoted as $m$, its actual values are reported in Table 2. We approximate the computation overhead of different baselines as follows.

1. **Static Extend**: $m \cdot (N + n)^2$
2. **Dynamic Extend**: $m \cdot N^2$
3. **Temporal Remask**: $(m + n) \cdot N^2$

4. **Concat Guidance**: $n \cdot n^2 + m \cdot N^2$

5. **Concat Latent**: $n \cdot n^2 + m \cdot N^2$

6. **LATTS**: $n \cdot n^2 + m \cdot N^2$

| Average answer length | GSM8K | MATH500 |
|---|---|---|
| Static Extend | 284 | 429 |
| Dynamic Extend | 286 | 561 |
| Temporal Remask | 284 | 401 |
| Concat Guidance | 238 | 328 |
| Concat Latent | 248 | 366 |
| LATTS | 284 | 399 |

Table 2: Average answer length of different sequential TTS methods.

## APPENDIX C  QUALITATIVE ANALYSIS OF SEMANTIC LATENT VECTORS

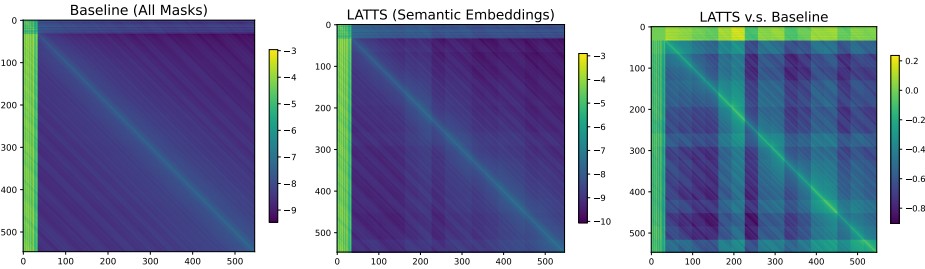

Figure 8: Attention Score analysis of LATTS v.s. baseline.

We conducted a comparative analysis of the initial attention patterns by using the masked embedding from the baseline method and the latent embedding from LATTS as starting points. For each case, we recorded the attention maps after the first layer, computed the geometric mean of the softmax-normalized attention scores across all heads, and then report the ratio of attention scores between the LATTS and baseline methods. According to Figure 8, two key observations emerge from this analysis:

Enhanced Prompt Attention: Tokens exhibit stronger attention to the prompt, as evidenced by lighter shading in the top rows of the attention map ratio visualization.

Strengthened Local Focus: The model shows increased local attention, a direct consequence of our chunk-level averaging and duplication strategy. This design results in similar initial values within the same block, thereby encouraging localized attention patterns.

Given the inherent sparsity of attention scores and consistent with prior sparse attention research Zaheer et al. (2020); Yuan et al. (2025); Zhang et al. (2023), which identifies critical regions such as the prompt, select global information, and local windows, our findings indicate that LATTS effectively enhances the model's utilization of contextual information.

# APPENDIX D  CASE STUDY

---

**(Question)** Amy had two eyeshadow palettes with four colors each and three makeup sets that came with six eyeshadow colors each. Her sister steals one of the palettes. Amy uses up half of the colors from one makeup set. How many eyeshadow colors does she have left?

**(Guided Answer)** Amy has 2 x 4 = 8 + 3 x 6 = 18 eyes. She has 8 - 1 = 7 + 6 / 2 = 10 eyeshadow colors left. Final Answer: The final answer is \boxed{10}

**(Concat Guidance)** Let's break down the problem step by step

1. Initial Count: Amy has two eyeshadow palettes, each with four colors. Total colors from palettes: 2x4=8. Amy has three makeup sets, each with six colors. Total colors from makeup sets: 3 x 6= 18. Total initial eyeshadow colors: 8 + 18 = 26.
2. Palettes Stolen: Her sister steals one of the palettes. Number of remaining palettes: 2 - 1 = 1. Total colors from remaining palettes: 1 x 4 = 4.
3. Palettes Used: Amy uses up half of the colors from one makeup set. Number of colors used from one makeup set: \frac{6}{2} = 3. Number of colors remaining from one makeup set: 6 - 3 = 3. Total colors from remaining makeup sets: 2 x 3 = 6.
4. Total Colors Remaining: Total colors from remaining palettes: 4. Total colors from remaining makeup sets: 6 Total eyeshadow colors remaining: 4 + 6 = 10
Final Answer: The final answer is \boxed{10}

**(Dynamic Extend)** To determine how many eyeshadow colors Amy has left, we need to follow these steps:
Step 1: Calculate the total number of eyeshadow colors Amy initially had. Amy had two eyeshadow palettes with four colors each. She also had three makeup sets with six colors each. Total colors from palettes: 2 x 4 = 8. Total colors from makeup sets: 3 x 6 = 18. Total initial number of colors: 8 + 18 = 26.
Step 2: Subtract the number of colors from the palettes that were stolen. One of the palettes was stolen, so the number of palettes left is 2 - 1 = 1. The number of colors from the stolen palette is 1 x 4 = 4. Total colors from palettes after the theft: 8 - 4 = 4.
Step 3: Subtract the number of colors from the makeup set that were used. Amy used up half of the colors from one makeup set. The number of colors in one makeup set is 6. Half of 6 is \frac{6}{2} = 3. Total colors from the makeup set after usage: 6 - 3 = 3. Now, add the colors from the palettes and the makeup set to find the total number of colors left. Total number of colors left: 4 + 3 = 7. Therefore, the number of eyeshadow colors Amy has left is \boxed{7}

**(LATTS)** To determine how many eyeshadow colors Amy has left, we need to follow these.
Step 1: Calculate the total number of eyeshadow colors Amy initially had. Amy initially had two eyeshadow palettes with four colors each and three makeup sets with six colors each.2 palettes x 4 colors/palette = 8 colors; 3 makeup sets x 6 colors/set = 18 colors; 8 colors + 18 colors = 26 colors
Step 2: Subtract the number of colors from the palettes that were stolen. Amy's sister stole one of the palettes, so she only has one palette left. 1 palette x 4 colors/palette = 4 colors; 26 colors - 4 colors = 22 colors
Step 3: Subtract the number of colors from the makeup set that were used. Amy used up half of the colors from one makeup set. 1 makeup set x 6 colors/set= 6 colors ;\frac{1}{2} x 6 colors = 3 colors; 22 colors - 3 colors = 19 colors.
Therefore, the number of eyeshadow colors Amy has left is \boxed{19}.

---

**(Question)** Richard wants to make a video to share online of him doing a science experiment that creates a fountain of diet soda after placing a specific branded candy inside it. Richard's driveway is 24 feet wide and he wants to put a bottle of soda every 3 feet of the driveway. After starting at the first bottle, it will take Richard 5 seconds to go from one soda bottle to the next, dropping the candy in. How many seconds total will it take Richard to set off all the soda fountains?

**(Guided Answer)** The are 24/3 = 8 bottles bottles of soda along the driveway. It takes Richard 5 seconds to go from one bottle to the next, so it will take 8*5 = 40 seconds. Final Answer: The final answer is \boxed{40}

**(Concat Guidance)** Step 1: Calculate the number of soda bottles along the driveway. The driveway is 24 feet wide, and Richard wants to place a bottle of soda every 3 feet. Therefore, the number of bottles is:

$$\frac{24 \text{ feet}}{3 \text{ feet/bottle}} = 8 \text{ bottles}$$

Step 2: Determine the time it takes to move from one bottle to the next. It takes Richard 5 seconds to move from one bottle to the next.
Step 3: Multiply the number of bottles by the time to move between bottles to find the total time. The total time to set off all the soda fountains is:

$$8 \text{ bottles} \times 5 \text{ seconds/bottle} = 40 \text{ seconds}$$

Final Answer: The final answer is \boxed{40}.

**(Dynamic Extend)** Step 1: Calculate the number of soda bottles. The driveway is 24 feet wide, and Richard wants to place a bottle every 3 feet. Therefore, the number of bottles needed is:

$$\frac{24 \text{ feet}}{3 \text{ feet/bottle}} = 8 \text{ bottles}$$

Step 2: Determine the time it takes to move from one bottle to the next. It takes Richard 5 seconds to move from one bottle to the next.
Step 3: Calculate the total time required to move through all the bottles. Since there are 8 bottles and it takes 5 seconds to move from one bottle to the next, the total time required is:

$$8 \text{ bottles} \times 5 \text{ seconds/bottle} = 40 \text{ seconds}$$

Thus, the total time it will take Richard to set off all the soda fountains is \boxed{40} seconds.

**(LATTS)** Step 1: Calculate the number of soda bottles. The driveway is 24 feet wide, and Richard wants to place a bottle every 3 feet. Therefore, the number of bottles needed is:

$$\frac{24 \text{ feet}}{3 \text{ feet/bottle}} = 8 \text{ bottles}$$

Step 2: Determine the time it takes to move from one bottle to the next. It takes Richard 5 seconds to move from one bottle to the next.
Step 3: Calculate the total time required to move between all the bottles. Since there are 8 bottles, Richard will need to move through 7 intervals between the bottles. The total time required is:

$$7 \text{ intervals} \times 5 \text{ seconds/interval} = 35 \text{ seconds}$$

Thus, the total time it will take Richard to set off all the soda fountains is \boxed{35} seconds.

---

We present a case study comparing LATTS with other sequential TTS algorithms. The figure illustrates the question along with the outputs generated by Concat Guidance, Dynamic Extend, and LATTS.

While Concat Guidance produces intermediate reasoning steps that are relatively detailed, its final output is identical to the short guided answer (length 64), which may be attributed to explicit error accumulation

introduced by spatial Chain-of-Thought. The Dynamic Extend method yields the longest final output. It adopts the overall idea of calculating the total by subtracting the stolen and used eyeshadow palettes, but implements it in an accumulative manner. Since the generation proceeds strictly left-to-right, errors cannot be effectively corrected once they occur, leading to an incorrect final result.

In contrast, LATTS produces the correct answer based on the draft answer, demonstrating its ability to perform error correction through additional steps in the latent space. This demonstrates that temporal CoT-based TTS leverages the self-correcting potential of DLLMs.

### D.1 CASE STUDY ON TEMPORAL REMASK

---

**(Question)** The set of vectors $\mathbf{v}$ such that

$$\text{proj}_{\begin{pmatrix}2\\1\end{pmatrix}} \mathbf{v} = \begin{pmatrix}0\\0\end{pmatrix}$$

lie on a line. Enter the equation of this line in the form "$y = mx + b$".

**(Origin)** Let $\mathbf{v} = \begin{pmatrix}x\\y\end{pmatrix}$. Then

$$\text{proj}_{\begin{pmatrix}2\\1\end{pmatrix}} \mathbf{v} = \frac{\begin{pmatrix}x\\y\end{pmatrix} \cdot \begin{pmatrix}2\\1\end{pmatrix}}{\begin{pmatrix}2\\1\end{pmatrix} \cdot \begin{pmatrix}2\\1\end{pmatrix}} \begin{pmatrix}2\\1\end{pmatrix} = \frac{2x+y}{5} \begin{pmatrix}2\\1\end{pmatrix} = \begin{pmatrix}\frac{4x+y}{5}\\\frac{2x+y}{5}\end{pmatrix}.$$

We want this to be $\begin{pmatrix}0\\0\end{pmatrix}$, so

$$\frac{4x+y}{5} = 0 \implies 4x+y = 0 \implies y = -4x.$$

The equation of the line is \boxed{y = -4x}.

**(Remask)** Let $\mathbf{v} = \begin{pmatrix}x\\y\end{pmatrix}$. Then

$$\text{proj}_{\begin{pmatrix}2\\1\end{pmatrix}} \mathbf{v} = \frac{\begin{pmatrix}x\\y\end{pmatrix} \cdot \begin{pmatrix}2\\1\end{pmatrix}}{\begin{pmatrix}2\\1\end{pmatrix} \cdot \begin{pmatrix}2\\1\end{pmatrix}} \begin{pmatrix}2\\1\end{pmatrix} = \frac{2x+y}{5} \begin{pmatrix}2\\1\end{pmatrix} = \begin{pmatrix}\frac{4x+y}{5}\\\frac{2x+y}{5}\end{pmatrix}.$$

We want this to be $\begin{pmatrix}0\\0\end{pmatrix}$, so

$$\frac{2x+y}{5} = 0 \implies 2x+y = 0 \implies y = -2x.$$

The equation of the line is \boxed{y = -2x}.

---

**(Question)** In the diagram, $D$ and $E$ are the midpoints of $\overline{AB}$ and $\overline{BC}$ respectively. Determine the area

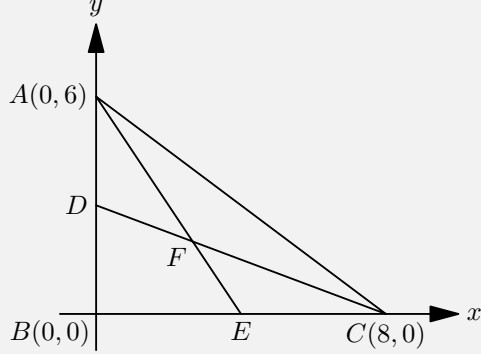

of quadrilateral $DBEF$. (The graph is drawn with asy.)
**(Origin)** The area of quadrilateral $DBEF$ is \boxed{8}.
**(Remask)** 8

|            | GSM8K | | MATH500 | |
|------------|-------|-------|-------|-------|
| $n$        | 256   | 512   | 256   | 512   |
| Remask(64) | 1.08  | 0.98  | 2.30  | 1.71  |
| ReMDM(64)  | 3.26  | 5.79  | 5.73  | 10.92 |
| ReMDM(512) | 2.96  | 5.65  | 5.76  | 10.11 |

Table 3: Average modified tokens of ReMDM and Remask

## APPENDIX E    ADDITIONAL EXPERIMENTAL RESULTS

### E.1    ADAPTING BEAM SEARCH TO DLLMs

We introduce a detailed description of implementing beam search for DLLMs. A challenge exists because the order of unmasked tokens is random in DLLMs. Our approach is based on the work of Vijayakumar et al. (2018), with modifications to fit the DLLM framework. The algorithm operates as follows:

1. **Initialization:** The $k$ beams are initialized as $k$ separate sequences. These beams are divided into $G$ groups, where each group initially contains $B = k/G$ beams.

2. **Denoising Step:** At each denoising step, the following operations are performed:

   - **Token Unmasking:** To enable effective diversity promotion in DLLMs, it is essential that the unmasking position remains fixed across all groups. The unmasking position is determined by the beam with the highest joint probability among all groups. This ensures that the sequence with the highest overall probability at that point guides the unmasking process.

   - For the $i$-th group, the following operations are performed:

     - **Token Scoring:** Each beam in the $i$-th group computes the log-probabilities of the candidate tokens based on the current partial sequence.
     - **Diversity Promotion:** For each candidate token, if the token has already been selected by any of the previous $i - 1$ groups at the current decoding step, a fixed diversity strength is subtracted from its log-probability for each occurrence in those groups. This strength, denoted as $\alpha$, encourages the exploration of alternative tokens by discouraging the selection of tokens that have already been favored by other groups.
     - **Token Selection:** After the strength is applied, the $i$-th group selects the top $B$ candidate beams based on the adjusted log-probabilities. The joint probability of each beam is the sum of the adjusted log-probabilities for all tokens in that beam. These $B$ beams with the highest joint probabilities are selected for the next decoding step.

3. **Output Selection:** After denoising is complete, the sequence with the highest overall joint probability is selected from the $k$ final beams.

In our experiments, we set $G = k$, which means that each beam forms its own group. The diversity strength $\alpha$ was set to $0.5$. This approach ensures that the generated beams exhibit greater diversity in token selections while maintaining high-quality sequences.

## E.2 ABLATION ON DRAFT ANSWER LENGTH

|       | GSM8K |      | MATH500 |      |
|-------|-------|------|---------|------|
| $n$   | 256   | 512  | 256     | 512  |
| 64    | 79.5  | 84.7 | 37.2    | 41.6 |
| 128   | 77.8  | 84.5 | 37.4    | 40.8 |
| 256   | 80.0  | 78.0 | 37.2    | 38.4 |

Table 4: Ablation on the draft answer length $n$.

We directly employ the model trained with self-sampling using a draft answer length of 64, and extend the draft answer length during testing. The results in Table 4 indicate that increasing the length of draft answers can enhance model performance under certain conditions. This improvement may be attributed to the fact that longer answers contain more informational content, thereby yielding greater precision. However, as discussed in Section 4.2, more precise answers may potentially constrain the model's ability to escape local optima. We may further explore adapting to different draft lengths during the training phase.

## E.3 ABLATION ON BLOCK SIZE

We further evaluate LLaDA-8B-Instruct under different block size settings besides of block size 32. According to Figure 9, LATTS delivers strong performance and remains its generalization in a wide range of block sizes. The results show that LATTS is robust to variations in block sizes, suggesting its applicability in diverse scenarios with different optimal block sizes.

| Block Size | Method | GSM8K 256 | GSM8K 512 | MATH500 256 | MATH500 512 |
|------------|--------|-----------|-----------|-------------|-------------|
| 16         | SFT    | 77.5      | 83.9      | 35.0        | 40.4        |
|            | LATTS  | 80.2      | 84.2      | 36.8        | 40.0        |
| 32         | SFT    | 75.4      | 82.8      | 35.6        | 37.0        |
|            | LATTS  | 79.5      | 84.7      | 37.2        | 41.8        |
| 128        | SFT    | 66.8      | 77.9      | 30.6        | 37.0        |
|            | LATTS  | 74.0      | 80.1      | 32.8        | 37.6        |

Figure 9: LLaDA-8B-Instruct performance on different block sizes

## E.4 ADDITIONAL ABLATION ON DENOISING STEPS

To evaluate the sensitivity of LATTS to its draft generation, we measured final answer quality on MBPP at 256 tokens across draft lengths of 32, 64, and 128 while varying the denoising steps. Results indicate that performance is robust to this hyperparameter, as the number of steps has minimal impact on the output. Furthermore, we observe that the optimal number of denoising steps is inversely related to the draft length, with longer drafts requiring fewer steps to peak.

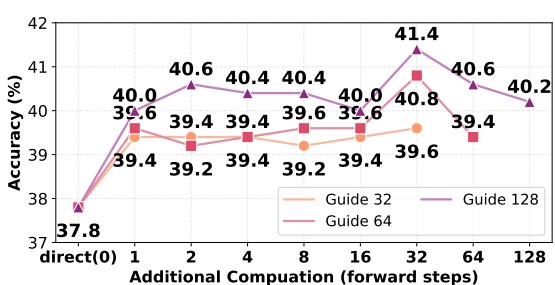

Figure 10: Denoising Step Sweep on MBPP.

## APPENDIX F    ELABORATION ON LATTS IMPLEMENTATION

We demonstrate the decoding process of LATTS, using the average embedding Aggregator $\mathcal{A}_{avg}$ as an example. Please refer to our released codebase for implementation details.

---

**Algorithm 1** LATTS Decoding Process

---

**Require:** Model $p_\theta(\cdot)$, word embedding $\phi(\cdot)$, answer length $N$, draft answer length $n$, prompt $c$
  1: // Differences to standard MDLM sampling noted in Green.
  2: **Phase 1: Construct Semantic Latent Starting Point**
  3:     $draft \leftarrow p_\theta(\cdot|c, n)$
  4:     $\mathbf{e}_{draft} \leftarrow \phi(draft)$                                   ▷ Map $draft$ to latent space.
  5:     $m = \lceil \frac{n}{L} \rceil, L' = \frac{N}{m}$
  6:
  7: **for** $i = 1$ to $m$ **do**                                   ▷ **Aggregation:** Compute chunk-level averages.
  8:         $chunk_i \leftarrow \mathrm{Avg}(e_{draft}[i * L : (i+1) * L])$
  9:     Let $chunks \leftarrow [chunk_1, chunk_2, \ldots, chunk_m]$
 10:     $semantic\_latent \leftarrow \mathrm{expand}(chunks, L')$           ▷ **Extension:** Duplicate each chunk $L'$ times
 11:
 12: **Phase 2: Decoding**
 13:     $S \leftarrow \{\mathbf{MASK}\}_N$                                   ▷ Initialize a fully masked sequence.
 14: **while 1**$(S == \mathbf{MASK}) > 0$ **do**
 15:         $latent = \begin{cases} semantic\_latent[i] & \text{if } S[i] = \mathbf{MASK} \\ \phi(S[i]) & \text{otherwise} \end{cases}$   ▷ Replace masks with semantic latent vector
 16:         $S \leftarrow p_\theta(|c, latent)$                        ▷ Perform denoising step according to Equation (2)
 17: **return** $S$

---

