# OpenReview forum: "LATTS: LAtent space Test Time Scaling for diffusion language models"
_ICLR.cc/2026/Conference — Submitted to ICLR 2026_

### Official Review · Reviewer_bbqK · 2025-10-29

**Soundness:** 1
**Presentation:** 2
**Contribution:** 2
**Rating:** 2
**Confidence:** 4

**Summary:**

The paper introduces LATTS, a method for test-time scaling in masked diffusion language models such as LLaDA. LATTS works by initializing and operating in the latent embedding space, enabling iterative self-refinement over a fixed-length sequence. This approach leads to notable gains over standard supervised fine-tuning baselines.

**Strengths:**

1. The experimental setup for training, and comparing LATTS and SFT is fair, the authors use the same training duration and hyperparameters.
2. On all reported tasks, LATTS improves the performance over regular SFT.

**Weaknesses:**

### Major weaknesses
1. The sampling procedure after initialization with LATTS is unclear. Are you using standard ancestral sampling starting from the LATTS embeddings? Do you decode intermediate tokens, or do you operate entirely in latent space without explicit decoding? Please clarify the exact sampling steps after initialization.
2. **Framing**: LATTS is described as a test-time scaling (TTS) method, but has two fundamental differences with regular test-time scaling approaches.
    - The authors *do not* increase the computational resources allocated to sampling, in order to measure test-time scaling. Instead, the authors argue that LATTS is as expensive as standard ancestral sampling. The "Temporal CoT" metaphor is also misleading: LATTS does not add extra denoising or refinement steps to answer generation, unlike regular CoT reasoning in AR models.
    - Secondly, unlike standard TTS methods, which use a fixed base model and require no fine-tuning, LATTS involves additional fine-tuning. This distinction is unclear in the abstract and introduction and may mislead readers about the actual contribution.
- The paper makes several claims that are not adequately supported by experimental evidence. For example:
    - Lines 195-196: The authors seek embeddings that are *"semantically rich and sufficiently abstract"*, but provide no evidence or analysis to support this.
    - On lines 82-83, the authors claim that LATTS *"mitigates the error propagation in spatial CoT and fully leverages the self-correction capabilities of DLLMs."*. I did not find evidence of self-correction or mitigation of error propagation in this submission.
- The paper contains several inconsistent statements:
    - The LATTS sampling process, is self-contradictory. First, LATTS produces $n$ draft tokens and aggregates their embeddings using a fixed operation like a mean, into a single vector, with the stated goal of avoiding starting sampling too close to a low-energy region (lines 204-205). Then, on lines 223-224, this vector is copied $K$ times (for $K$ mask tokens) to form the initial matrix, and the authors claim this shifts the starting point closer to a low-energy region. This is confusing: are you trying to avoid or approach the low-energy region in initialization? Please clarify the intended design and the reasoning.
    - Relatedly, lines 232-233 claim that the initialized embeddings *"implicitly encode token-wise similarities."* However, since lines 239-240 state that all embeddings are created by duplicating the same vector, how can they meaningfully represent token-wise similarities? This needs clarification.

### Other weaknesses
1. The abstract inaccurately claims dLLMs require fixed-length generation. Prior work, such as Llada supports variable and semi-autoregressive generation, *which is investigated in this paper in Table 8*.
2. **Limited selection of downstream tasks**. Thanks to LLM evaluation frameworks, such as lm-eval-harness, it should be straightforward to compare LATTS and regular ancestral sampling on multiple downstream tasks, while you only report Arc-challenge and HellaSwag. This is not sufficient and could raise concerns whether the results of LATTS were poor on other tasks.

- The claim that the number of refinement steps is limited by sequence length (abstract) only applies to ancestral sampling. Remasking methods like ReMDM [1] allow more sampling steps than tokens by reintroducing masks.

- Key dLLM works are missing from the citations: D3PM, MDLM, MD4, and RADD.

- Lines 237-238: Please avoid calling your strategy "optimal" without evidence; you have not shown there is no better alternative.

[1] Remasking Discrete Diffusion Models with Inference-Time Scaling, Wang et al.

**Questions:**

1. While you report the hyper-parameters for fine-tuning with LATTS and regular SFT, you do not compare the actuall wall-time duration, and throughput of LATTS training and SFT.
2. Similarly, what is the impact of LATTS on sampling speed? As stated in the weaknesses, it is not clear to me from the paper how sampling is implement, besides the initialization of tokens. The caption of Figure 3 states *"estimated inference time computation (lower better)"*, however it is not clear what units of measurement is used
3. On lines (274-275), you state the the length of the draft is always 50% of the total sequence length. Why this choice? What happens if you use 10%, 25%, 75% of the total tokens as draft?
4. On line 281, you say that you use the standard lm_eval framework. Are you using lm-eval-harness? If so, can you cite it? Note that lm-eval-harness was designed for autoregressive models, in which case, how are you adapting it for diffusion language models?
5. The "Temporal Remask" baseline performs poorly, which serves as a key justification for LATTS's more complex initialization strategy. However, this baseline remasks tokens randomly. How would LATTS compare to a stronger baseline that remasks tokens based on the lowest confidence scores from an initial generation?

---

> ### Author Response · Authors · 2025-11-25
> **Response to Major Weaknesses**
>
> We thank the reviewer for their thorough review and constructive suggestions. We apologize for any lack of clarity in our original description of the LATTS algorithm and provide the following clarifications.
>
> **Regarding Major Weakness 1: Clarification of the LATTS Algorithm**
>
> We sincerely apologize for the unclear presentation. We added detailed explanations (Section 4.1) and pseudocode (Appendix E) in the article. The actual procedure is as follows:
>
> - **Step 1:** Sample the semantic latent embeddings (which remain fixed throughout the subsequent process).
> - **Step 2:** Replace the embeddings of all mask tokens with these semantic latent embeddings. Then, compute the logits for all tokens, and decode the top-k tokens with the highest confidence (i.e., the "intermediate tokens" mentioned). This step closely resembles traditional decoding, with the key difference being that we replace the embeddings of still-masked tokens with their corresponding semantic latent embeddings, while unmasked tokens retain their own embeddings.
>
> We loop **Step 2** until all tokens are unmasked.
>
> **Regarding Major Weakness 2: Computational Cost and Training Overhead**
>
> - **Inference Cost:** We did not state that LATTS has the same cost as traditional decoding, only that it introduces low inference-time overhead. The abstract mentions "minimal additional inference computation," and Section 5.2 repeatedly discusses "added computation," with Figure 3 providing a detailed overhead analysis. The primary overhead comes from acquiring the semantic latent embeddings in Step 1. Since the draft is short, this cost is modest. The subsequent Step 2 incurs costs comparable to standard decoding. In the revised version, we directly report the end-to-end inference latency (please kindly refer to **General Response 1**), showing LATTS introduces only **5-10% latency overhead** and achieves an almost Pareto improvement over baseline methods, proving its cost efficiency.
>
>     Concerning the "Temporal CoT" metaphor, we argue that Step 1, which performs information aggregation within the model, can be seen as an extension of the denoising process. The aggregated embedding may provide helpful hints for the decode stage (see **General Response 2** for details).
>
> - **Training Overhead:** We did not conceal that LATTS requires additional post-training. Our original Contribution 3 in the introduction explicitly stated this, and we have now also mentioned it in the abstract of the revised version. This post-training is a single-time effort to teach the model to utilize the latent embedding as a starting point for iteration. Compared to the more extensive pre-training required for models like LLaDA-8B (which used 2.3 trillion tokens and 0.13 million H800 GPU hours, as reported in their paper), our overhead is relatively minor (please refer to our response to **Question 1** for specific numbers).
>
> **Regarding Major Weakness 3: Experimental Support for Claims**
>
> We have revised the relevant claims in the manuscript for greater precision. Furthermore, we have made the following efforts to substantiate them:
>
> - **"Semantically rich and sufficiently abstract":**
>     - *Semantically Rich:* We provide two pieces of corroborating evidence:
>         1. In **General Response 2**, we visualize the impact of our embeddings on attention maps, showing they guide tokens to attend to more informative parts of the context.
>         2. Our experiments integrating LATTS with parallel TTS (Figure 4) demonstrate that starting from this initial point increases the probability of multiple decoding trajectories converging to the correct answer, indicating the embedding is semantically rich.
>     - *Sufficiently Abstract:* Figure 5 shows that increasing denoising steps on the draft does not monotonically improve accuracy; peak performance often occurs with fewer steps. We empirically interpret this as suggesting that overly specific information is not necessarily beneficial. Additionally, Figure 6(b), where we vary the chunk size for aggregation, reveals a performance peak, indicating that either too fine-grained or too coarse-grained chunks are suboptimal.
> - **"Mitigates error propagation...":** We have rephrased this claim in the revision to the more measured: **"yields fewer cascading errors and a higher final accuracy, as evidenced by our experimental results."**
> Our failure mode analysis (Appendix C) suggests that because our method iterates in the latent space, it is less prone to directly copying erroneous answers from previous steps. We have added another case study to illustrate this. In contrast, we observed that baseline methods like Dynamic Extend and ConcatGuidance more frequently produce errors due to a single mistake in an intermediate step.

---

> > ### Author Response · Authors · 2025-11-25
> > **Response to Major Weaknesses 2**
> >
> > **Clarification on Seemingly Inconsistent Statements**
> >
> > We would first like to clarify our aggregation and extension process. It does **not** involve simply duplicating an identical latent vector across all positions, as suggested. For the **Average Embedding Aggregator**, the actual implementation performs **chunk-wise duplication**. To be specific: for a draft of length 64, a block length `L` of 4, and a target answer length of 256, we first aggregate the draft into 64/4=16 latent vectors. Each of these 16 vectors is then duplicated 16 times (as 256 / 16 = 16) to form the final initial latent sequence.
> >
> > **(1) LATTS Decoding Process and Intuition**
> >
> > We apologize for any lack of clarity in explaining the decoding process and intuition behind LATTS. The two statements referenced are, in fact, **not contradictory** but complementary. Our underlying reasoning is as follows:
> >
> > The energy manifold of the solution space typically contains multiple local optima. The goal of our latent starting point is to be **closer to a low-energy region** than a fully masked initialization, yet **not so close to any specific local optimum** as to cause the "basin stickiness" problem during the subsequent denoising phase. This "stickiness" would prevent the model from escaping a sub-optimal basin even if a better solution exists nearby. This rationale shares the same motivation as the "semantically rich yet sufficiently abstract" property discussed earlier.
> >
> > Therefore:
> >
> > - The statement mentioned in lines 204-205 explains *why* we need aggregation and extension: to **blur the draft information**, preventing the initial point from being too close to a specific local optimum and thus mitigating basin stickiness. We provide empirical evidence for this problem in our response to Reviewer 3 weakness 4 (see Appendix D.1), showing that initializing directly with token embeddings (which are too specific) results in very few tokens being modified during denoising.
> > - The statement in lines 223-224 aims to convey that our latent starting point is **nevertheless closer to a low-energy region** compared to a fully masked initialization. This is experimentally validated in Figure 4, where we demonstrate that multiple decoding trajectories starting from this point have a higher probability of converging to the correct answer (i.e., a low-energy region).
> >
> > **(2) Clarification on Latent Starting Point Composition**
> >
> > Contrary to the impression received, our latent starting point is **not formed by duplicating a single, identical latent vector**. As detailed in the initial clarification, the process is one of **chunk-wise duplication**.
> > Consequently, the resulting initial sequence contains **groups of identical vectors**, which allows the model to compute similarity information with respect to the prompt and other blocks, functionally resembling a form of **block-based attention** in the initial latent space.

---

> ### Author Response · Authors · 2025-11-25
> **Responses to Other Weaknesses**
>
> 1. We thank the reviewer for this comment. We have revised our statement by adding "standard" and "without specific training" to reflect this point with greater precision.
> 2. For evaluation, we utilized the lm-eval-harness, following the implementation in LLaDA (https://github.com/ML-GSAI/LLaDA). Our initial reporting focused primarily on math, code, and commonsense tasks, partly because our fine-tuning dataset, s1K, is designed to enhance reasoning capabilities. We have now expanded our evaluation to include BBH, TruthfulQA (ROUGE-1 accuracy), and IFEval (instruction following) on LLaDA-8B-Instruct. LATTS demonstrates improvements over the SFT baseline across these tasks, with a particularly notable gain(+19.6%) on TruthfulQA.
> | method | BBH(512)| BBH(256) | TruthfulQA(512)| TruthfulQA(256)|IFEval(512)| IFEval(256)|
> |:---------------:|:---------------:|:---------------:|:---------------:|:---------------:|:---------------:|:---------------:|
> | direct    | 56.3    | 57.6    | 54.6   | 53.1 |63.3|	61.6|
> | sft   | 56.6	 | 57.9 | 50.4	 | 49.7 | 65.6|	64.6|
> | latts   | 57.0|	57.8	| 51.2	|69.3 |66.0	|64.9|
>
> 3. Thanks for this suggestion. We've made our expression more precise in the abstract (lines 18-21). We have also incorporated ReMDM and ALMD into the related work section. Furthermore, we adapted the ReMDM strategy on LLaDA (see Figure 3), LATTS outperforms ReMDM with different hyperparameters.
> 4. We sincerely appreciate this suggestion. We have now included discussions of D3PM, MDLM, MD4, and RADD in the related work section.
> 5. Thank you for this suggestion. We have rephrased our claims to be more professional. Additionally, we have introduced further baselines, such as replacing random remasking with low-confidence remasking and comparing against ReMDM. LATTS continues to maintain a performance advantage over these methods.

---

> ### Author Response · Authors · 2025-11-25
> **Response to Questions**
>
> 1. Thank you for this suggestion. We have now included a description of the training wall-clock time in the manuscript (see lines 306-308).
> 2. Please refer to **General Response 1** for a detailed discussion on this point.
> 3. We apologize for the misunderstanding. To clarify, please refer to our original statement in the paper: *“The number of denoising steps used when sampling the draft answer is set to be half of the draft answer length.”* This indicates that the draft is generated at a rate of two tokens per denoising step. We conducted an ablation study on the number of denoising steps for the draft in Figure 5 of the original paper, and an ablation on the draft length in Appendix D.2.
> 4. Please refer to our response in **Other Weakness #2**. We utilized the lm-eval-harness (which we have cited) and adapted it for DLLM evaluation by following the implementation in the LLaDA repository.
> 5. We thank the reviewer for this suggestion. We modified the original temporal remasking strategy from random to a lowest-confidence-based remasking. However, this modification did not yield significant performance gains and did not surpass LATTS. We hypothesize this is because once a token is unmasked, it is difficult for the remasking process to alter its value, as its associated confidence typically becomes very high. An analysis of the error cases for this method is provided in Appendix D.1. We also included the ReMDM baseline for comparison, which likewise did not outperform LATTS.

---

### Official Review · Reviewer_8UsB · 2025-10-31

**Soundness:** 3
**Presentation:** 2
**Contribution:** 3
**Rating:** 6
**Confidence:** 4

**Summary:**

The paper introduces LATTS, a test time scaling method for masked diffusion language models that reframes chain of thought from a spatial process, adding tokens, to a temporal process, adding refinement steps in latent space. Concretely, the model first self-samples a short draft answer, aggregates the draft’s token embeddings into a small set of semantic latent vectors, expands these vectors to the target length, mixes them with the mask embedding, and uses this as the new initial latent state for the standard denoise–remask procedure. The intuition is that a semantically informed starting point steers iterative refinement toward better basins in the embedding landscape. The method is simple to implement and requires only a brief post-training stage. Experiments show modest gains on LLaDA-8B across GSM8K, MATH500, MBPP, ARC-Challenge, and HellaSwag relative to supervised fine tuning.

**Strengths:**

1. The temporal chain of thought view is a clean conceptual bridge for diffusion LMs, and it is well integrated with the masked diffusion decoding loop. The paper clearly formalizes the remask rule and places the method in the latent optimization picture.
2. The two core components, aggregation of draft embeddings and blockwise expansion with mask mixing, are straightforward and easy to implement.
3. Timely contribution. Extending TTS to Diffusion LLMs isn’t a trivial task.

**Weaknesses:**

1. The reported improvements over supervised fine-tuning are in the three to five percent range, depending on task and length, which is modest at best.
2. All core results use LLaDA-8B Base and Instruct. There is no replication on an alternative diffusion LM. This makes it difficult to judge generality.
3. The potential energy narrative in Figure 2 is insightful but remains qualitative. There is no formal argument connecting the initialization to objective improvement or basin selection under the remask schedule.
4. The paper argues that naive temporal remasking changes very few tokens post-convergence, which hints at a basin “stickiness” phenomenon, but it does not quantify it across methods or give a more direct look at failure modes.
5. The illustrative figures are largely uninformative and fail to convey new insight beyond the text

**Questions:**

1. Can the authors move beyond the current heuristic chunking approach and explore sentence-aware or adaptive segmentation strategies that better align with semantic boundaries? Fixed-size windowing may misrepresent reasoning units within latent space. Recent work demonstrates that boundary-aware chunking substantially enhances retrieval and reasoning, including [1,2,3,4] have also established the benefits of linguistically informed segmentation. Citing and empirically engaging with these works would situate LATTS within the broader literature on semantic chunking and help demonstrate that its latent-space aggregation leverages meaningful linguistic structure rather than relying purely on heuristic pooling.
2. The paper’s generality claim would be much stronger if replicated on a second masked diffusion LM and at least one additional model size. Even a small scale experiment on another eight billion model, or a four billion model, would help.
3. The scaling plot shows diminishing returns as extra steps increase. Please extend the curve farther for at least one task and length to find the true plateau, and report the best operating point across lengths.
4. You sample drafts with half as many denoising steps as the draft length. How sensitive are results to the draft schedule and to the temperature used for the draft? Please include a table that varies draft length, draft steps, and sampling temperature while keeping total compute fixed, to see whether the best practice is to spend compute on a better draft or on more final refinement.

[1] Zhao, Jihao; Ji, Zhiyuan; Feng, Yuchen; Qi, Pengnian; Niu, Simin; Tang, Bo; Xiong, Feiyu; Li, Zhiyu. *Meta-Chunking: Learning Efficient Text Segmentation via Logical Perception.* arXiv preprint arXiv:2410.12788 (2024).
[2] Zhu, Yuxuan; Falahati, Ali; Yang, David H.; Mohammadi Amiri, Mohammad. *SentenceKV: Efficient LLM Inference via Sentence-Level Semantic KV Caching.* arXiv preprint arXiv:2504.00970 (2025).
[3] Duarte, André V.; Marques, João D. S.; Graça, Miguel; Freire, Miguel; Li, Lei; Oliveira, Arlindo L. *LumberChunker: Long-Form Narrative Document Segmentation.* Findings of EMNLP 2024 (2024).
[4] Sheng, Boheng; Yao, Jiacheng; Zhang, Meicong; He, Guoxiu. *Dynamic Chunking and Selection for Reading Comprehension of Ultra-Long Context in Large Language Models.* arXiv preprint arXiv:2506.00773 (2025).

---

> ### Author Response · Authors · 2025-11-25
>
> Thanks for your insightful comments, we have conducted the following experiment to address your concerns:
> 1. [To Q1] We appreciate this valuable question. The four cited works, which we have now added to Section 4.1 and mentioned in future work, focus on sentence-level chunking for long-text processing. Given that our drafts are short (64/128 tokens), a direct transfer is non-trivial.
> Our current solution to avoid semantic truncation is a punctuation-based (.,!?) sub-sentence chunking of the draft. We aggregate embeddings within each sub-sentence and then proportionally duplicate them to align with the final answer's length.
> Our evaluation below indicates the limited efficacy of this straightforward approach.
>  This is due to the insufficient number of segments generated from short draft answers, a problem that is compounded by the inherent non-robustness of variable-length chunking.
> We think this is an interesting future direction, but due to limited time and resources, it will be hard to integrate the PPL-based ("Meta-Chunking") and similarity-based ("Dynamic Chunking") methods into LATTS. We would do our best effort if requested.
> | method | gsm8k(256)| gsm8k(512) | math500(256)| math500(512)|
> |:---------------:|:---------------:|:---------------:|:---------------:|:---------------:|
> | sft    | 75.4    | 82.8    | 35.6    | 37.0 |
> | latts   | 79.5	 | 84.7	 | 37.2	 | 41.8 |
> | latts (vari-chunking)    | 77.5|	83.2	| 36.8|	37.2|
>
>
>
> 2. [To W2 & Q2] Thanks! We have added new experiments with the 2B LLADA model. Please refer to General Response 3. We will strive to include more models during rebuttal, updating our results as possible.
>
> 3. [To W3] We appreciate this rigorous question. We believe a post-hoc analysis of the results can shed light on how the initialization influences the selection of optimal attraction basins or leads to objective improvement. Please refer to General Response 2 for details.
>
> 4. [To W4] First, we refined the naive temporal remasking strategy into a low-confidence remasking approach. Furthermore, we have detailed in the Appendix D.1 the average number of tokens altered by this method across different tasks, supported by a case study of instances where the correction led to the right answer. Our analysis on the MATH500 dataset reveals that due to the low average number of token modifications, only 4 samples exhibited a change in their final answer after remasking. Specifically, among these, one was corrected due to an altered numerical value, one was altered incorrectly, and the remaining changes were primarily related to formatting.
> We also included the RemDM baseline, which employs an iterative remasking and unmasking process. While this looped mechanism introduces a greater number of token modifications, it also incurs significantly more computational overhead. Ultimately, its performance fails to surpass that of LATTS, as detailed in Figure 3.
>
> 5. [To Q3]  Thanks for the advice. We have supplemented our evaluation with experiments on MBPP, elaborated in Appendix E.4 . For draft lengths of 32, 64, and 128 tokens, we report the performance impact of varying the number of denoising steps on the final answer quality (measured for answers of length 256, which yields higher quality on this benchmark). Our results indicate that the number of denoising steps applied to the draft has a minimal effect on the final output. This finding demonstrates the robustness of LATTS to variations in draft quality. We observe that the optimal performance is achieved at a moderate denoising step, and this peak step is smaller for longer drafts. This finding directly corresponds to our discussion in Section 4.1: the draft embeddings must be semantically informative yet abstract.
>
> 6. [To Q4] We thank the reviewer for this question. Our initial investigation into the sensitivity of draft length and denoising steps is presented in Figure 5 and Appendix E.2, where the sampling temperature was fixed at 0. These results, along with our new comprehensive experiments on MBPP, collectively indicate that LATTS performance is generally robust to these hyperparameters.
> Regarding the specific request for a controlled-compute trade-off analysis between draft quality and final refinement, we acknowledge its value. In our current setup, we observed that the impact of LATTS on inference time is minimal. Consequently, within the range of this low overhead, the fluctuations in time and accuracy make it difficult to delineate a consistent optimal trade-off from the noise.
> Please note whether we need further investigation.

---

### Official Review · Reviewer_JXwA · 2025-11-01

**Soundness:** 2
**Presentation:** 3
**Contribution:** 2
**Rating:** 6
**Confidence:** 5

**Summary:**

This paper introduces LATTS (Latent space Test-Time Scaling), a novel method for improving the inference-time performance of diffusion large language models (DLLMs). The authors identify that conventional sequential Test-Time Scaling (TTS) methods, such as Chain-of-Thought (CoT), are incompatible with modern masked DLLMs. This incompatibility stems from two key constraints of DLLMs: they operate on fixed-length sequences and have a limited number of refinement steps, which prevents the dynamic token-level expansion required for CoT.

The core idea of LATTS is to reframe CoT from a "spatial" process (extending sequence length) to a "temporal" one (extending iterative refinement in the latent space). The method works in two stages:

1.	First, it uses the model to generate a short "draft answer".
2.	The embeddings of this draft are then aggregated (e.g., via average pooling) into a compact set of "semantic latent vectors".
3.	These vectors are then used to create a new, semantically-informed initial latent state for the final answer, replacing the standard all-mask starting point.

The model is briefly post-trained to learn how to effectively refine from this new starting point. Experiments on the LLaDA-Instruct model show that LATTS achieves notable gains on reasoning and code benchmarks (e.g., +4.1% on GSM8K, +4.8% on MATH, +3.2% on MBPP) with minimal additional computational overhead.

**Strengths:**

1.  Sound Solution: The paper's key conceptual contribution, the "Temporal Chain of Thought," is a novel, sound, and well-motivated solution to the problem of applying sequential reasoning to non-autoregressive, fixed-length DLLMs. The paper makes a solid and valuable contribution to the field by cleverly bypassing the architectural constraints of full-attention DLLMs.

2. Strong Empirical Results: The central claims are well-supported by the evidence. LATTS demonstrates consistent and significant performance gains over strong SFT baselines across multiple challenging benchmarks, including mathematical reasoning (GSM8K +4.1%, MATH +4.8%) and code generation (MBPP +3.2%).

3. Excellent Clarity and Presentation: The paper is very well-written, and the core concepts are explained with exceptional clarity. The introduction of "Spatial CoT" versus "Temporal CoT" is an intuitive and effective framing. The figures are high-quality and add significant value, and the paper structure is logical.

4. Thorough Experimental Setup: The experimental setup is generally sound. The method is tested on a range of reasoning, code, and commonsense tasks. The paper includes a strong comparative analysis against five other sequential TTS baselines (e.g., Static Extend, Concat Guidance, Temporal Remask) and shows that LATTS provides a Pareto-optimal improvement. The ablation studies on hyperparameters are also thorough.

**Weaknesses:**

1. Missing Latency Benchmarks: The paper's primary weakness is the complete omission of practical, end-to-end latency measurements. The analysis relies solely on theoretical computational estimates. This is insufficient for a method explicitly designed for test-time scaling, as it introduces an entire extra generation step (sampling the draft answer) whose real-world latency cost is never measured.
2. Limited Exploration of Decoding Strategies: The experiments appear confined to the default LLaDA decoding setup: a semi-AR, fixed-step process where the number of denoising steps is tied to the answer length. The paper fails to explore how LATTS interacts with the parallel decoding proposed by Fast-dLLM, which allows the model to generate the answer via variable denoising steps. As far as I can see, by providing a semantically rich starting point, LATTS could reduce the number of refinement steps needed to converge, potentially making the model faster end-to-end. It would be better if the authors could provide extended experiments on this issue.
3. Limited Scope (Full-Attention DLLMs): The paper's experiments and analysis are focused exclusively on LLaDA, a DLLM that uses full attention over the entire sequence at each denoising step. The authors do not discuss the applicability or potential modifications needed for LATTS to work with newer, more efficient block-diffusion models (e.g., D2F, SDAR, Fast-dLLM-v2), which are a growing trend in the field. This limits the paper's significance.
4. Questionable Baseline Implementation: The "Dynamic Extend" baseline results in Figure 3 are surprisingly strong. Many practitioners find that LLaDA suffers severe performance drops with this method, as they tend to terminate generation prematurely. The paper does not provide sufficient hyperparameter detail to explain its decent reported performance, raising minor concerns about the baseline's implementation.
5. Superficial Related Work on CoT: The paper positions LATTS against a "Spatial CoT" strawman (simple, linear generation). The "Related Work" section fails to discuss or compare against more advanced, non-linear CoT methods that also address efficiency and parallel generation, such as "Skeleton-of-Thought".
6. Lack of Qualitative Analysis of Latent Vectors: While Appendix C provides a case study, it only compares the final text outputs. The paper would be much stronger if it included a qualitative analysis of the semantic latent vectors ($a_j$) that form the core of the method, to provide intuition for why they are a good starting point.

**Questions:**

1.	Could the authors please provide end-to-end, wall-clock latency benchmarks (e.g., time per sequence) for the SFT baseline vs. LATTS? This is essential for understanding the practical performance-latency trade-off.
2.	Have the authors considered experimenting with the parallel-decoding strategy proposed by Fast-dLLM? It seems plausible that the "better starting point" provided by LATTS could allow the model to reach a high-quality answer in fewer denoising steps than the fixed number used in the experiments.
3.	The "Dynamic Extend" baseline results in Figure 3 are surprisingly strong. Given that LLaDA models often suffer from premature termination with this strategy, could the authors please elaborate on the exact hyperparameters and prompting used to achieve these decent baseline results?
4.	How does the "Temporal CoT" paradigm conceptually compare to other advanced CoT methods for DLLMs such as "Skeleton-of-Thought"?
5.	The LATTS method seems intrinsically tied to the full-attention, holistic refinement of LLaDA. How could this method be adapted to work with newer block-diffusion DLLMs (like D2F or SDAR)?
6.	Could the authors provide a more in-depth, qualitative analysis of the semantic latent vectors ($a_j$) themselves? A visualization, for example, could provide valuable intuition as to how these aggregated draft embeddings create a more effective starting point.

---

> ### Author Response · Authors · 2025-11-25
>
> Thank you for your constructive feedback. We have made the following revisions to address your points:
>
> 1. [To W1 & Q1] Regarding Weakness 1 & Question 1, please refer to **General Response 1**, where we have provided detailed latency data.
> 2. [To W2 & Q2] Thank you for this suggestion. Concerning the experimentation with various decoding strategies, we have incorporated an additional *dynamic*, *confidence-based* decoding method, inspired by Fast-dllm. This method decodes a token if its confidence exceeds a predefined threshold. We set the confidence ratio to 0.9, meaning that tokens with confidence greater than 0.9 are decoded at each step. We report the accuracy and per-token average latency for answer lengths of 256 and 512 on the GSM8K and MATH datasets. During the decoding process of LATTS, the sampling of draft answers follows the exact same setup as described in Section 5.1. Under the confidence-based approach, we observe that LATTS achieves significant accuracy improvements compared to the baseline SFT (conf) method, while the accuracy degradation relative to the default LATTS is not substantial. Moreover, it delivers up to 4.4× acceleration compared to the default LATTS in decoding speed.
> | method | GSM8K(512) | GSM8K(256) | MATH500(512)| MATH500(256)|
> |:---------------:|:---------------:|:---------------:|:---------------:|:---------------:|
> ||Acc(%) / Latency(s)|Acc(%) / Latency(s)|Acc(%) / Latency(s)|Acc(%) / Latency(s)|
> | sft (conf) | 82.3 /  4.88 | 76.9  /  3.26  | 38.0  /  7.66 | 35.6  / 4.37 |
> | latts  | 84.7 /  25.78 | 79.5  /  11.58  | 41.8 / 25.99 |  37.2  /  11.64 |
> | latts (conf) | 83.9 /  5.87	| 78.0	/  4.26 | 40.8	/  8.02 | 36.7 /  5.28 |
> 3. [To W3 & Q5]  We sincerely appreciate this insight. In principle, LATTS can generate a shorter answer that retains complete semantic information by constraining the answer length, and then use this as the latent starting point for subsequent iterations. However, adapting this mechanism to the Block Diffusion framework is non-trivial. Block Diffusion employs an autoregressive (AR) approach to generate blocks and uses diffusion to create the intra-block content. Integrating LATTS into the generation process *within* each block presents a specific challenge: since Block Diffusion is trained on fixed block sizes, it is difficult to generate a draft latent embedding of a *shorter* length that still encapsulates the complete information for that specific block.
>
>     Given current time and resource constraints, we will make our best efforts to explore this adaptation with Block Diffusion and report any findings in future work.
>
> 4. [To W4 & Q3]  Regarding the implementation of our dynamic extension mechanism, we have provided further clarification in Section 5.2.1. Assume the answer length is N, block length is m. The process initializes a sequence with `N` masked tokens. During semi-autoregressive decoding, after a block of m tokens are unmasked, a new fully-masked block of the m is appended to the end if no EOS token is generated. Generation stops only upon EOS detection or when the maximum length (`2N`) is reached. This design prevents premature termination because the model's context window at each step remains `N` tokens, ensuring it is never cut off mid-generation. The purpose of this baseline is to prevent the model's output from being artificially constrained by a pre-defined fixed length. The corresponding code is available in our repository under `evaluate.py evaluate_baseline_extend → generate_extend`.
> 5. [To W5 & Q4] We appreciate the reviewer's comment. We have cited and discussed SoT in the related work. Skeleton of Thought (SoT) and our LATTS both follow a high-level “draft-then-refine" structure. Applying SoT directly to DLLMs faces two key issues: (1) Its efficacy diminishes on logically dense tasks (like math or code) that resist clear multi-step decomposition, as noted in SoT's own evaluation (Sec. 3.2.3). (2) Its parallel expansion is incompatible with DLLMs' requirement for bidirectional attention. The variable and uncertain length of parallel segments prevents the application of global attention, making mutual awareness between concurrently generated points impossible.
> 6. [To W6 & Q6] Thanks for pointing this out. Please refer to our **General Response 2**.

---

### Official Review · Reviewer_6aRU · 2025-11-03

**Soundness:** 2
**Presentation:** 3
**Contribution:** 3
**Rating:** 6
**Confidence:** 4

**Summary:**

Test-time scaling (TTS) enhances autoregressive models by allowing additional computation during inference. However, existing sequential TTS approaches do not directly apply to diffusion models because:
(1) diffusion models operate on fixed-length sequences, and
(2) the number of iterative refinement steps is capped by the sequence length.

This paper addresses these challenges by reframing chain-of-thought reasoning from a *spatial* process (extending sequence length) to a *temporal* process (adding refinement steps during denoising). Experimental results show an average improvement of **4.6%** on GSM8K, MATH, and MBPP benchmarks, with minimal additional inference cost.

**Strengths:**

1. The proposed sequential test-time scaling for diffusion large language models improves reasoning performance over base diffusion models. Importantly, the performance after applying TTS becomes comparable to methods trained with DPO variants, while requiring **99% less training data**.

2. As shown in **Table 1**, LATTS outperforms both the base and SFT models while using significantly less computation.

3. The ablation study in **Figure 3 (left)** clearly demonstrates the advantage of LATTS over dynamically extending the sequence during semi-autoregressive decoding, which represents the strongest baseline among the compared methods.

**Weaknesses:**

1. **Lines 18 & 64–66: “refinement steps capped by sequence length”:**
   This claim is not accurate. Prior DLLMs (e.g., ReMDM: https://openreview.net/pdf?id=xNwZ8kDC7T, ADLM: https://openreview.net/pdf?id=E8adS5srds) show that increasing denoising steps **beyond** the sequence length improves generation quality. ADLM further demonstrates that with sufficiently many steps, DLLMs can surpass strong AR models and even larger DLLMs. These works should be discussed in Related Work, and the statement should be revised.

2. Is optimizing **only the initial embeddings** sufficient before running the standard reverse diffusion? What is the wall-clock cost of this optimization, and how does it scale with model size? Could this optimization be applied (or updated) at **every time step**, and if not, why?

3. The method appears to average draft word embeddings and then proceed with standard denoising. Even if random remasking provides little benefit once tokens are unmasked (Fig. 3 left; Lines 368–369), why can’t **LATTS be applied at the end** (i.e., after full unmasking) as a final refinement?

4. The paper claims to address (1) fixed-length sequences and (2) capped iterative refinement. However, the algorithm still assumes a **fixed answer length** and a **fixed cutoff** (e.g., 2048) augmented with averaged draft embeddings. In what precise sense does LATTS relax the fixed-length constraint, beyond increasing effective refinement steps via initialization?

**Questions:**

Please see the weakness section above.

---

> ### Author Response · Authors · 2025-11-25
>
> Thank you for this constructive feedback. We would like to provide the following clarifications and modifications:
>
> 1. [To W1] We apologize for overlooking the two referenced works. We have made the following revisions to address this:
>     a) We have refined our description regarding the "refinement steps capped by sequence length" in the abstract and introduction to present a more precise statement (see lines 18-21, lines 65-69).
>     b) We have included and discussed both works in our related work section. Please refer to lines 118-120.
>     c) We have included comparative experiments with RemDM in our evaluation. Our experimental setup follows the configuration adapted for LLaDA in the original RemDM paper. We evaluated the performance of ReMDM across different numbers of looped steps. Our results demonstrate that LATTS consistently outperforms ReMDM across the evaluated trade-off space.
>
> 2. [To W2] Thank you for raising these three points. We address them individually below:
> (1) Our current experimental results demonstrate that optimizing only the initial embedding already yields notable improvements over baseline methods. As visualized in General Response 2, LATTS's latent starting point enables more effective information extraction throughout the decoding process. We believe further refinement strategies represent a promising direction for future work.
> (2) Regarding wall-clock time, as detailed in General Response 1, LATTS introduces minimal overhead and remains efficient among the most baseline methods. For ablation studies concerning model size, please refer to General Response 3.
> (3) Your suggested approach is indeed intriguing. We implemented a variant where, after unmasking tokens at each step, we replace corresponding positions in the initial semantic latent embedding with the embeddings of decoded tokens, followed by re-applying the Aggregate and Extend operation to update the semantic latent embedding for subsequent mask positions.
> | method | gsm8k(256)| gsm8k(512) | math500(256)| math500(512)|
> |:---------------:|:---------------:|:---------------:|:---------------:|:---------------:|
> | sft    | 75.4    | 82.8    | 35.6    | 37.0 |
> | latts   | 79.5	 | 84.7	 | 37.2	 | 41.8 |
> | latts (update)    | 77.5| 	81.7	| 36	| 37.8|
>
> However, empirical results indicate this method underperforms.
> We hypothesize this occurs because the initial semantic latent embedding already contains comprehensive, albeit fuzzy, information about the entire block. Progressively replacing it with decoded embeddings disproportionately amplifies attention to already-decoded tokens. Particularly in later stages, when most tokens are unmasked, the masked positions become contaminated with excessive information from unmasked tokens, thereby impairing the model's ability to focus effectively on the remaining masked content.
>
> 3. [To W3] Thank you for the question. We want to ensure we address the part we didn't fully understand. Are you proposing that after standard decoding of all tokens, we apply LATTS to create a latent embedding, then remask some tokens (by random or confidence) and refine using this new embedding as the initial state? Please feel free to clarify, and we can run follow-up experiments as needed.
> 4. [To W4] Thank you for raising this point. We would like to clarify that our work does not claim to solve variable-length generation in DLLMs. When we refer to "fixed length" in the abstract and introduction, our intention is to highlight that naive DLMs cannot freely extend sequence length spatially, and thus cannot directly support spatial Chain-of-Thought (CoT) expansion like AR models can. This limitation motivates the need to extend the denoising steps temporally, which we term temporal CoT. LATTS operates within a fixed output length but extends the thinking/refinement process along the temporal (iterative) dimension, not the spatial one. We did not intend to suggest that our method addresses variable-length generation.

---

### Author Response · Authors · 2025-11-25
**General Responses**

We sincerely thank all reviewers for their thorough and constructive feedback. We have identified several common concerns and addressed them collectively below:

**(1) Decoding Latency Comparison**

We have added Figure 3, which directly reports the per-sample decoding latency of LATTS compared to other TTS methods. All measurements were taken on an H200 GPU with a batch size of 1. Our comparisons show that LATTS introduces only 5-10% additional overhead relative to the standard AR baseline. This makes LATTS one of the most efficient TTS methods while simultaneously achieving the highest accuracy improvement. Please refer to Section 5.2.1 for a detailed analysis.

**(2) Quantitative Analysis of Latent Vector Effectiveness via Attention Maps**

In Appendix C, we quantitatively evaluate the effectiveness of the initial latent vectors produced by LATTS. Specifically, we extract the initial mask embedding (default) and the LATTS-generated latent vector, then compute the geometric mean of the softmax-normalized attention maps after the first layer. We plot the ratio of attention scores between the two settings. Our findings indicate that tokens attend more strongly to the prompt (evidenced by lighter shading in the top rows) and exhibit enhanced local attention. Given that attention scores are inherently sparse, and prior work on sparse attention has identified key regions, such as the prompt, certain global information, and local windows, we conclude that LATTS strengthens the model’s utilization of contextual information.

An alternative way to measure effectiveness is to perform multiple sampling runs from the same initial point and evaluate the likelihood of converging to the correct answer. This is equivalent to decoding multiple trajectories using both the baseline and LATTS-enhanced parallel TTS, which we have already discussed in Figure 4.

**(3) Ablation Study with a 2B LLaDA Model**

In our ablation studies, we have trained and incorporated a 2B-parameter LLaDA model. Due to time and resource constraints, and considering that no other parameter-sized LLaDA models have been released, we trained this 2B model from scratch to serve as a base for experimentation. The results, detailed in Figure 7, demonstrate that LATTS also achieves consistent improvements at this smaller scale.

---

### Author Response · Authors · 2025-12-02
**Summary of Revisions**

We sincerely thank all reviewers and the Area Chair for your time and insightful feedback. Below, we summarize the overall improvements we have made in response:

(1) **Added Evaluation**: We conducted end-to-end comparisons of LATTS’s report latency against various baseline methods in terms of wall-clock latency (Figure 3), demonstrating that LATTS achieves Pareto improvements in both performance and efficiency.

(2)  **Interpretability of LATTS**: We performed a quantitative analysis (Appendix C) of the draft embeddings obtained by LATTS and found that the improved starting point enhances both information extraction and attention, further validating its effectiveness.

(3)  **Scalability on other model scales**: Since no other publicly available parameter sizes exist for LLADA, we trained a 2B LLADA model ourselves. We added Figure 7 to report the performance, and LATTS consistently maintained its advantages.

(4)  **Expanded baselines and downstream tasks**: We strengthened the temporal remask baseline with low confidence remasking and added ReMDM (an enhanced temporal remark baseline in the literature), and observed that LATTS still stands out across all baselines in terms of both efficiency and accuracy. Additionally, we incorporated three distinct benchmarks: BBH, TruthfulQA, and IFEval, and found that LATTS improves over SFT by up to 19.6%.

(5)  **Implementation and evaluation of reviewer‑suggested variants**: We implemented and tested several LATTS variants proposed by the reviewers, including: updating the semantic latent vector prior at each decoding step (*reviewer 6aRU* W2); employing variable‑length chunking during aggregation (*reviewer 8UsB* Q1); and adapting to Fast‑DLLM (*reviewer JXwA* W2/Q2).

(6)  **Improved writing quality and added references**: We refined several arguments to be more precise, added Section 4.1 to further clarify the algorithm flow, and included algorithm pseudocode in Appendix E. All references mentioned by the reviewers have been thoroughly discussed and appropriately cited.

For each specific point raised by the reviewers, we have provided detailed revisions in the corresponding response sections. Please refer to the individual responses for further details.

---

### Meta-Review · Area_Chair_eGs5 · 2026-01-08

**Summary:**

This paper proposes a "test time scaling" (TTS) method for diffusion language models. It finetunes a dLLM by first generating a draft answer, which is then transformed into an informative starting state for the sampling process. Experiments on LlaDA shows that this consistently outperforms the SFT baseline on a range of benchmarks.

Three reviewers are in favor of accept while one reviewer has concerns. After going through the rebuttal, the AC believes that it boils down to the central claim: whether the proposed approach is indeed test time scaling. On this point, the AC believes that reviewer bbqK's weakness 2 made some fair points: standard TTS has the benefit of yielding increasingly better results by dynamically expanding its sequence length, whereas the proposed approach is more like adding a constant computation to the pretrained model to increase its power. The AC does believe that this distinction needs to be made clear and upfront to avoid confusions. Although this is nonetheless a very promising piece of work as pointed out by other reviewers, the AC cannot recommend accept at this point.

**Reviewer Concerns:**

See summary, main issue is weakness 2 from Reviewer bbqK.

**Reviewer Scores:**

I'd expect reviewer bbqK to remain negative but others remain positive.

---

### Decision · Program_Chairs · 2026-01-26

Reject